# Post-Pandemic Lessons for Destination Resilience and Sustainable Event Management: The Complex Learning Destination

**Jesse Carswell** [1], **Tazim Jamal** [2] , **Seunghoon Lee** [2,*] , **Donna Lee Sullins** [2] and **Kelly Wellman** [1]

1   Office of Sustainability, Texas A&M University, College Station, TX 77843-2261, USA
2   Department of Recreation, Park and Tourism Sciences, Texas A&M University,
    College Station, TX 77843-2261, USA
*   Correspondence: shlee@tamu.edu

**Abstract:** This paper aims to share post-pandemic lessons for destination resilience and the sustainability of events. It offers a new perspective that reimagines the space and place of events as *learning destinations* enmeshed in complex systems. Complexity arises due to the interactions and interrelationships between numerous stakeholders, activities, and events in the social–ecological destination system, where boundaries are porous, and issues and actions from afar can impact the local community. The case presented here describes the micro-level activities and actions undertaken to engage with destination resilience and sustainable event management and certification at a learning destination in Texas, USA. These situated efforts are shown (i) at the campus-wide level for the university and (ii) with the collaborative, learning-oriented activities undertaken by students in event management classes to pilot test the Sustainable Event Certification Checklist that was developed. They corroborate the general characteristics and criteria of the complex learning destination summarized in the paper, along with identifying and discussing the skills, literacies, and lessons learned to advance destination resilience and the sustainability of events. Participants in the learning destination draw on practical knowledge and develop soft skills to engage in adaptive planning proactively and collaboratively with other stakeholders to address emergent challenges and practical problems in the complex destination and sustainable event domain.

**Keywords:** learning destination; adaptive planning; destination resilience; stakeholder collaboration; sustainable event certification

## 1. Introduction

Event tourism is one of the fastest-growing sectors worldwide, offering considerable economic and social benefits locally and globally through festivals and events, e.g., music festivals and sporting events, including mega-events such as Coachella, the Olympics, or the World Cup. Pre-pandemic annual participation in business events, for instance, was around 1.5 billion visitors worldwide, contributing USD 1.5 trillion to global GDP, with direct and indirect spending constituting USD 2.5 trillion (Ref. [1] cited in [2]). Leisure spending by tourists at festivals and events adds significantly to economic impacts and is highly beneficial to local communities, generating wealth through employment, tax revenues, and diverse business opportunities (see, for example, Ref. [3]).

However, events suffer from the innate disadvantage of high vulnerability to internal or external disturbances, such as extreme weather and environmental impacts, biosecurity risks, domestic as well as international terrorism, and incidents such as the deadliest mass shooting in American history that occurred on 1 October 2018 at a country music concert in Las Vegas, which left 58 dead and over 600 injured. Over half the concert attendees were estimated to be from California, and federal funding to support them and first responders from outside the state helped to supplement heavily stretched state resources [4]. More

recently, the prolonged impact on events and destinations overall due to COVID-19 and its severity has been unparalleled compared to other disease outbreaks and coronaviruses such as SARS and MERS [5,6].

Research on the pandemic's impacts is still in its early stages. Greater attention will be needed to manage events in the context of infectious disease outbreaks as online to offline life resumes. However, online and virtual activities will continue to be essential aspects of event planning and marketing [7]. This Special Issue calls for reimagining how events are to be designed, implemented, and evaluated in the future and reassessing their value to people, organizations, and destinations. Miles and Shipway (2020) called for studies on how international sports events are affected, and both sport and event management studies can be better informed by disaster management and resilience studies [8]. They also note the need for more attention to synergizing wider social, community, and individual resilience perspectives. Recent disasters such as the crowd crush incidents at Astroworld in Houston, Texas, USA, and during Halloween in Seoul, Korea [9], add further urgency to identify strategies and ways to increase resiliency to local and globally driven disasters. Destination resiliency is therefore the focus of our paper, as the context in which complex environmental, social and economic impacts associated with events arise and synergistically influence the ability of the destination to withstand disasters and adverse impacts.

Events are intricately embedded in place and context. They are affected by multiple local issues within the destination (e.g., the socioeconomic and environmental impacts of event tourism on the place) in addition to global factors (climate change, pandemics, etc.). A collaborative approach to learning how to address such challenges is needed. No stakeholder alone can resolve the complex issues and wicked problems that may emerge spontaneously and unpredictably in the event domain. Our purpose here is to offer a new perspective that reimagines the space and place of events as learning destinations [10] that strive for resilience and sustainability within complex social–ecological systems. Complexity arises due to the interactions and interrelationships between numerous stakeholders, activities, and events in the destination system, where boundaries are porous. Consequently, issues and actions from afar can impact the locals. Situated, place-based collaboration facilitates knowledge and resource sharing between stakeholders, enabling adaptation as well as event policies that can contribute to broader regional development and sustainability goals [11,12].

Hence, this paper offers a holistic, place-based approach to post-pandemic sustainability and resilience, where learning, adaptive planning, and collaborative engagement are critical factors. Micro-level actions undertaken by an academic educational institution toward overall destination resilience and the environmental and social sustainability of local events provide a practical case example that corroborates the proposed approach of a learning destination [10] (see also [13]). Here, stakeholders jointly learn and dynamically adapt to situations and emergent challenges in that particular place. Hudson (2013), cited in [10] (p. 340), in fact, argued that academia should be a significant contributor to the process in the form of being a "bridge builder" among stakeholders [14]. Sadd et al. (2017) follow up by saying, "Despite the exponential growth in tourism academia in many countries around the world it is surprising that Hudson notes that such 'bridge building' remains rare with academics not fully leveraging their knowledge and skill set to the benefit of destinations" [10] (p. 340).

The main aims of this paper are therefore to (1) make a conceptual contribution to the notion of a learning destination, aided by a literature review that helped to identify important dimensions for building resilience, and (2) present a case study that corroborates the learning destination criteria identified and further strengthens the framework by identifying important competencies for learning destinations striving for resilience and sustainable event development, planning and management. The main research questions we address here are as follows: How can the notion of a learning destination be conceptually and practically strengthened for building destination resilience? How can the

learning destination approach help to guide sustainable event development, planning, and management?

Sadd et al. (2017) introduced the notion of the learning destination and used a case study approach to forward strategic directions for event planning and evaluation in the learning destination [10]. We have similarly adopted a case study approach that offers valuable lessons for destination resilience and sustainable event management. The case example of the learning destination provided here is that of Texas A&M University. It is a public university tasked with facilitating knowledge and learning, the public good and well-being. It is an educational and recreational destination, hosting numerous sports and social and cultural events. Actions and initiatives toward destination sustainability and resilience include developing a sustainable event certification process.

The following section commences with a brief review of the impacts of events and impact management considerations, followed by some situated approaches for building destination resilience. The key characteristics of the learning destination are then identified, which sets the context for the case study presented in the subsequent section. Post-pandemic lessons for destination resilience and the sustainability of events are then discussed, followed by conclusions and directions for future research and practice.

## 2. Literature Review

### 2.1. Event-Related Impacts and Impact Management

Destinations experienced significant adverse economic and social impacts from the loss of tourism and event revenues during the COVID-19 pandemic. Fortunately, social media and disruptive technologies enabled swift transition and exponential growth in online and virtual events, aided by design innovations and the use of marketing, promotion and communication tools (QR codes, YouTube live streams, online booking platforms etc.). Governments stepped in to help the event sector worldwide, recognizing its vital importance in sustaining local and regional economies and its social and psychological health value as social isolation progressed in 2020 and 2021. Outdoor events in many destinations received government support, recognizing the importance of offering economic and social resilience and their potential to become super-spreading sources of infection and outbreaks [2]. In addition, financial and policy support was vital to enable proper guidelines and measures to address crowding, social distancing, masking, etc., as well as developing new marketing and promotion strategies and online presence and engagement [15].

However, numerous other issues and impacts arise in enacting and managing events. Studies of the impacts of festivals and events indicate that resource conservation, ecological considerations, and the conservation of the natural environment rank highly among concerns expressed by local residents. Exploratory factor analysis of the impacts of major sports events, for instance, revealed four positive outcome areas: image and status; international exchange and cooperation; economic and tourism development; and infrastructure development. In addition, adverse issues related to inconveniences in daily life, environmental pollution, and security concerns were also identified [16]. Transportation is critical in infrastructure considerations, and its environmental impacts are significant. Bottrill et al. (2009), cited in [17] (p. 265), state that "if a group of people are encouraged to congregate in a particular location by a specific event, the event organizers should be responsible for the movement and implementation of strategies to reduce the negative impact of transportation to a venue" [18]. Event venues tend to consider transportation to and from venues as an indirect emission and generally do not consider this a responsibility. However, as Musgrave (2011) notes, some do plan to avoid unnecessary emissions in transit to events and set sanctions, such as Liverpool Arena and Convention Centre, Manchester Evening News Arena, and the O2 in London [17].

Research on the perceived legacies of international events such as the Olympic Games shows that residents perceive environmental legacies as the most important across cities and over time, followed by economic and sociocultural legacies [19]. Recognizing the adverse environmental impacts of mega-events, the Sydney 2000 Games offered a compre-

hensive environmental plan, consisting of 100 commitments, covering the five key areas of energy conservation, water conservation, waste avoidance and minimization, pollution management, and the protection of significant natural and cultural environments [20].

The UN Global Compact (2008) provides ten widely accepted principles on labor and human rights, the natural environment, and corruption to guide businesses that could help strengthen corporate social responsibility through responsible and sustainable business policies and practices [21]. These could be especially valuable for international and large-scale mega-events that often use migrant labor to build infrastructure and could be drawn upon as needed in the local context of destination resilience, where events play an integral part in ecological sustainability and community well-being.

A diverse range of social impacts also arises at place-based events: the influx of tourists, urban regeneration, community pride, entertainment value, improved reputation, community image, community attachment, employment opportunities, and sociocultural benefits (see [22–24]). Social interaction and favorable relationships between visitors and local residents can generate a welcoming atmosphere and help to promote local culture and traditions [25]. However, qualitative research by Wilmink-Thomas (2021) also shows, among other things, how consumer demand for sustainable festivals is assumed to change: either visitors will expect more due to growing awareness and abundant time to get informed, or the desire to celebrate rompishly will suppress considerations of sustainability and place resilience [26].

Not surprisingly, environmental responsibility is intricately interwoven with social responsibility within the destination context in which events arise. Key stakeholders, ranging from local businesses, residents, and visitors to non-profit organizations and various public sector participants, engage collaboratively to assist in greening events, sustaining the destination, and facilitating place resilience, including its inhabitants' well-being [27]. Ahmad et al.'s (2013) research indicates that communication and awareness raising play key roles in facilitating learning and knowledge transfer in critical dimensions for creating more environmentally sustainable events, which they identify as energy efficiency, waste minimization, water consumption, and eco-procurement [27].

### 2.2. A Holistic and Situated Approach

In the study of event management and design, many common elements traditionally cited for success are understanding the purpose, planning for the client's needs, researching, recruiting and assessing vendors, aesthetic design, risk management, site design, staff training, and management [28,29]. Over the last decade, event textbooks have provided a new commentary on understanding and implementing sustainability practices in event design and implementation. They highlight the need for emerging and experienced event managers to prioritize sustainability as integral to successful events and destination resilience. A review of the research literature shows that this involves, among other things, taking a holistic, situated approach to event management that jointly addresses local as well as global issues, such as climate change and conserving the natural environment [20].

Examining short-duration festivals from the perspective of sustainable tourism development, McKercher et al. (2006) similarly emphasized the need for the event and the destination to grow and develop together; the environmental and social impacts of events must be incorporated within a more holistic, integrated approach to the sustainability of place and events [30]. Such an approach is also advocated by Getz (2009), who notes, "Sustainable events are not just those that can endure indefinitely, they are also events fulfilling important social, cultural, economic, and environmental roles that people value. In this way, they can become institutions that are permanently supported in a community or nation. Green events are part of this movement" [31] (p. 70). In addition, as Mair and Laing (2012) note, greening events can be an excellent opportunity to educate and change attitudes and behaviors within the organization and toward the natural environment [32].

Learning and awareness raising play important roles in this context. Yuan (2013) notes that sustainable event management requires careful planning prior to events, and

offering information and education to raise the awareness of event goers is key to putting sustainability into practice [33]. Exploring future trends and issues in greening events, Frost, Mair, and Laing (2014) identify three educational opportunities: to raise awareness, to encourage behavior change as part of a more extensive campaign, and to use the festival or event to play an advocacy role [34]. They observe that events such as the Manchester Festival in the UK pride themselves on reaching out to the local community, making them a part of the green celebration whether the community members physically attend the events or not. Such pro-environmental and community-oriented action is integral to event and destination resilience.

A Circular Economy

Following from the above, a holistic approach to destination resilience sustains the local community through a place-based, situated perspective that integrates nature-positive ecological, social and cultural values within a circular economy [35]. A circular economy is a green and healthy economy where sustainable development and well-being are achievable within the limits of the ecological carrying capacity of the social–ecological system that constitutes the destination and the events hosted within it. It rejects the established linear economic model of production and consumption that assumes an unlimited supply of natural resources and an infinite environmental capacity to absorb waste. Applying the circular economy approach to tourism and events in the Anthropocene encourages striving to reduce a destination's carbon footprint and facilitate sustainable approaches, such as organic regenerative agriculture and sourcing local foods and place-based activities offered by service providers that engage in responsible, green practices. It also addresses the tourist experience, seeking ways to encourage pro-environmental behavior and reduce waste through enjoyable participation (e.g., at local recreational and cultural events), enabling "good actions toward conservation, encouraging civic responsibility and much-needed policies for conservation of local to global natural areas of visitation" [35] (p. 434).

For example, Bluesfest, Byron Bay, Australia, is a festival that respects, transforms, and grows the local community; it aims toward being a zero-waste festival with carbon-neutral initiatives, serving foods that are fair trade and organic, addressing waste disposal policies and facilitating a circular economy: it is a festival site that is a functioning, healthy and natural ecosystem [34,36].

*2.3. A Learning-Based Paradigm: The Complex Learning Destination*

The holistic, integrated approach above is vital to addressing the numerous impacts and issues in the event management domain. Building sustainability and resilience in the event-based destination also requires learning, adapting, and managing multiple stakeholders and the dynamic context in which issues arise. Sadd et al. (2017) introduce the concept of the learning destination that we draw on here as a valuable paradigm. It is an "inclusive, holistic and collaborative stakeholder entity that joins together in continuously shaping the future direction of destinations through the collective sharing of perspectives and knowledge for the greater good of the destination" [10] (p. 346).

As they mention, the term "learning destination" was inspired by a similar concept in the management literature: the learning organization. "Defined by Senge (1990) as an organization where "people continually expand their capacity to create results they truly desire, where new and expansive patterns of thinking are nurtured, where collective aspiration is set free, and where people are continually learning how to learn together, [37] (p. 3)" the learning organization sits within the wider, and expanding "knowledge" ecosystem" [10] (p. 340). Continuous learning (life-long learning) and knowledge sharing through inclusion and collaboration among salient stakeholders are crucial at both the organizational and destination levels in order to provide adaptive responses, say Sadd et al. (2017) [10].

However, we emphasize that the learning destination is situated within a complex social–ecological system, and attending to the context, things, and events in the system is essential if destination resilience and sustainability are to be accomplished effectively.

Attractions such as events are not isolated; they are part of the complex destination system. Their sustainability and resilience are interrelated with the emergent and dynamic properties of the complex system and the situated practices of multiple stakeholders within the local system and external to it [38]. Therefore, strategies for sustainable event management should be tailored to the place rather than aspire to universal norms, as Dredge and Whitford (2010) note [39]. They can be guided by some general principles along with specific principles tailored to the place and context.

The stakeholders (including local residents) of a complex learning destination system engage in adaptive planning and management to address emergent problems and wicked problems such as climate change, as well as engaged learning, collaborative engagement, and a holistic, situated approach to destination resilience within a circular economy. It requires engaging practical wisdom (*phronêsis* in the Aristotelean sense) and empathy to see the perspectives of other stakeholders and grapple with unpredictable, newly arising environmental and social issues from within or external to the destination (e.g., extreme weather and sea-level rise due to climate change, biosecurity and terrorism, disease outbreaks, and pandemics).

It also requires making trade-offs and decisions that may only check some of the boxes in a sustainability framework. Greening is about progress, not perfection. Ahmad et al. (2013) note that sustainability does not happen with just a single event [27]. Every decision and action presents opportunities for collaborative learning, building soft skills and knowledge: it is a process of continual improvement and innovation [17,27].

Discussing event management skills in the post-COVID-19 World, Werner, Junek, and Wang (2022) drew on qualitative, semi-structured interviews to explore the viewpoints on requisite future skills from three groups of event stakeholders, professionals, lecturers, and students, across three countries: China, Germany, and Australia [40]. The results indicate that the most important event management skills will be technical and digital expertise (e.g., to help with the increasing significance of digital and hybrid event forms), communication, innovation, and leadership skills, along with other skills, such as risk and crisis management skills, critical thinking and problem-solving skills, teamwork and collaboration skills, and soft skills around adaptability, agility, dealing with uncertainty, flexibility, openness to change, patience, and resilience. The professionals interviewed also pointed to the need for practical skills (applied knowledge) and stronger cooperation between universities and enterprises [40].

### 2.4. Building Adaptive Resilience through Collaborative, Proactive Planning

In the context of events, Sadd et al. (2017) state that the learning destination is "a vehicle to develop an inclusive and strategic approach to event planning that facilitates organizational learning and more effective decision making at the destination level" [10] (p. 346). Some important steps they describe are vital stakeholders coming together to identify objectives, determine evaluation criteria, place weightings and test criteria and the overall framework with event stakeholders and within events, tweak as needed, and soft-launch the framework for new events. In their paper, Sadd et al. (2017) use this more holistic approach to facilitate the development of an event portfolio to enhance and build tourism and enable adaptation through collaborative learning and knowledge sharing at the destination level [10].

Stakeholder involvement is crucial in planning and implementing green events. Laing and Frost (2010) emphasize involving key stakeholders in planning green events and developing policies and practices about waste management, recycling, traffic control, food miles, fair trade, carbon offset, and so on [36]. As noted earlier, visitors are also important stakeholders and should be involved through awareness raising and engagement in event sustainability. Wong et al. (2015) examined green policies and practices and assessed attendees' willingness to pay for them with the help of a green involvement scale they developed [41]. Scale items addressed sustainable food, green design and waste management, green activity and energy use, and a healthy environment.

In addition to stakeholders, two areas that need to be emphasized in the post-COVID-19 learning destination are adaptation and appropriate evaluation criteria. Liu-Lastres and Cahyanto (2021) drew on lessons learned during the COVID-19 pandemic to identify the importance of proactive (rather than reactive) planning and building adaptive resilience into strategic planning in order to build a resilient event industry [42]. Adaptive resilience can be greatly aided by technology and innovation. In the context of sporting events, for instance, Bazzanella et al. (2021) observe that the computerization and digitization of most organizational procedures (registration management, accreditation, delivery of race packages, etc.) will continue to increase [43]. Such virtual adaptations are integral to the post-COVID paradigm shift to facilitate event sustainability and destination resilience. These authors advocate creating new and specific international standards for managing event competition sites in the event of a new pandemic. As they discuss, standardized rules and their timely application could allow events to be carried out safely anywhere in the world while managing the extent of infectious disease breakouts [43]. As the alignment of safety, financial, and sustainability goals converges, their dominance in practice can find greater support and success.

Surprisingly, despite its immense local to global significance and importance in the context of recreation, leisure, and tourism, sustainability-related accreditation schemes for the event industry still need to catch up. In the absence of standardized rules and benchmarks, as well as accreditations, best practice guides and a variety of approaches to conducting greening festivals and understanding the key elements required to ensure success have arisen [17,44]. However, certifications for individual professionals in sustainable event planning have been gaining traction (see globalgreenevents.org and https://www.mpi.org/education/certificate-programs/sustainable-event-strategist, accessed on 9 January 2023). Moise and Macovei (2014) have argued that national and international organizations and institutions should be established to certify the right of the event organizers to promote their events as green events if they meet certain standards regarding aspects such as transport, catering, energy consumption, waste disposal, innovation, and ways of broadcasting the event [45]. Furthermore, different green levels and logos can also be created so that the audience can identify the ecological level that the event obtained [45].

### 3. Building Resilience in the Learning Destination

The above review of the research literature provides some valuable insights into the characteristics of learning destinations for event management and the processes and skills needed to address the complex, interrelated, local to global scope of impacts affecting the event domain, as well as the importance of proactive planning and preparation to manage crises such as future disease outbreaks when they occur. Addressing event- and place-based sustainability and building destination resilience are important goals for all event providers. Successful events contribute to ecological and social well-being in the destination, in addition to the equitable distribution of the economic benefits (and costs) of the event. Drawing from the above review, dimensions of particular importance to building resilience in the complex learning destination are shown in Figure 1.

Appropriate guidelines, frameworks, and actions for sustainable/green event management and destination resilience are important to develop but may vary depending on the context. Among other things, they are influenced by place-based needs (environmental and social), stakeholder priorities, and public–private sector and political agendas. The following section describes the micro-level activities and actions undertaken to engage with destination resilience and sustainable event management at a learning destination in Texas, USA. These situated efforts are shown (i) at the campus-wide level for the university and (ii) in relation to the activities undertaken by students enrolled in the event management classes involved. These efforts were practically oriented and not driven theoretically by the notion of the learning destination. However, the case illustrates and corroborates many of the criteria in the bulleted list above and offers lessons for building sustainable events and

resilience through adaptive learning and collaboration. The activities undertaken by these students will be described in greater detail below.

- **Collaborative engagement** among key stakeholders, sharing knowledge and learning, joint decision making, etc.

- **Developing soft skills and exercising practical knowledge** (practical wisdom to make ethical and fair decisions on sustainability and trade-offs as required, deal with ambiguity and emergent issues in complex system)

- **Creativity in designing "solutions" to wicked problems** with empathy to understanding how problems are affecting others in the event destination domain

- **Technological skills** to address online and onsite event marketing, etc.

- **Impact identification and sustainable event management skills** (being especially aware of the interrelatedness, emergence, and long-term effects on ecological, social and cultural sustainability and well-being; drawing on the circular economy, etc.)

- **Awareness raising and educational skills** to inform and engage diverse visitors and local stakeholders to contribute to destination resilience and sustainability of place and events

- **Proactive and adaptive planning and disaster preparedness** (policies, training, strategies, etc.)

- **Event sustainability policies, frameworks and certification, etc.,** to help guide event development, management, and marketing in the complex destination system

**Figure 1.** Some important dimensions for building resilience in the complex learning destination.

## 4. The Texas A&M Learning Destination: Building Resilience and Event Sustainability

Texas A&M University (TAMU) is the state's first public institution of higher learning and one of the largest Tier One research institutions in North America. The College Station campus serves 68,461 undergraduate and graduate students. TAMU is a destination for student learning and professional development and a hub for numerous recreation activities, including sports, cultural, and community event destinations.

Kyle Field, home of the Texas A&M football team, seats 102,733 fans and generally hosts 6–7 home games annually. Reed Arena seats 12,989 and is home to men's and women's basketball and volleyball teams. Reed Arena also serves as an important special event center. It has become a leading destination for neighboring school districts within 60 miles to hold high school graduation.

TAMU brings the world to the doorstep of the Bryan-College Station community. The Bush School of Government & Public Service brings global leaders in areas of international affairs, political science, and national security and intelligence to campus every semester. The Memorial Student Center Opera and Performing Arts Society (MSC OPAS) presents professional productions of theater, music, and dance programs that enrich the lives of the local community. With over 1100 student-led organizations, all of which host multiple meetings and events annually, students are at the forefront of many event-based decisions in our campus community.

### 4.1. Case Study Approach

The case study described below follows a service-learning and experiential education approach. It is a collaborative, place-based approach to sustainability and resilience in the learning destination, where students and multiple other stakeholders jointly engage in decision making, learning and knowledge transfer, and organizational, social, and institutional change (see Refs. [46,47]). As Chupp et al. (2010, p. 190) propose, "intentionally aiming for impact at three levels—on students, on the academic institution, and on the community—may be the key to making the most of any service-learning project [48]." They, too, used a case-based service-learning research approach to show how students, faculty,

and universities as a whole can benefit as they reach out to engage with community members and jointly facilitate meaningful change. The redistribution of decision-making power, more equitable and mutually beneficial relationships between students and community members, skill building, knowledge sharing, and learning are additional important benefits of service-learning and experiential education approaches [49], as the case study below demonstrates.

In this case, the Office of Sustainability staff (including a student intern), the instructor, and students in a total of five event management classes in 2021 and 2022, as well as a range of community stakeholders, including event producers and venue managers as described below, were key stakeholders. The event management classes comprised a total of 86 students in Fall 2021, 67 in Spring 2022, and 24 in Fall 2022. Each worked with the sustainable event framework draft being developed by the OS. They were closely guided by their instructor, who kept observation notes, regularly summarized accomplishments, and took photos of students engaged in event planning and implementation. The sustainability element of production for this student project was not new, but was greatly expanded upon with the use of the TAMU Sustainable Event Certification Checklist, which served as not only a reflective tool but also an instructive one for students seeking to improve the sustainability of their events and needing direction in how to do so. Incentives of both academic assessment by the instructor and, in some cases, financial assistance (see further below) ensured that students' efforts in sustainability were prioritized in their event plans.

As illustrated below, some important benefits and outcomes of the service-learning and experiential education approach are synergistic knowledge transfer and skill acquisition, building important competencies and literacies through engaged learning and multi-stakeholder collaboration, as well as concrete outcomes for destination resilience and sustainable event development, planning, and management (see also [50,51] cited in [52]). The benefits accrue not only to the students (and faculty) but also to the university and wider community (the event management community and overall learning destination resilience, as described and discussed below).

### 4.2. Efforts to Address Sustainability

The university is an essential focal point for education, recreation, and entertainment with a significant environmental impact. As such, TAMU looks for ways to minimize its impact on natural resources and lower greenhouse gas emissions. The campus Sustainability Master Plan (SMP) provides a blueprint to take measurable, attainable actions for the university. It focuses on 9 themes and defines 16 evergreen goals and 47 targets over a 20-year timeline. The plan is organized into four focus areas: physical environment, waste management, social sustainability, and institutional effort.

One of the key SMP evergreen goals is to achieve carbon neutrality by 2050. TAMU's primary greenhouse gas reduction effort is to reduce the overall campus energy consumption. For the baseline year (FY04), the total campus energy consumption was 6.5 million mmBtu (1 Million British Thermal Units) for a campus size of 20.5 million gross square feet (GSF). In the measurement year (FY19), the total campus energy consumption was 5.5 million mmBtu for a campus size of over 29.6 million GSF. Even though the campus saw tremendous growth, the overall energy consumption was reduced by 15%. This reduction in energy consumption was accomplished with a new combined heat and power plant, a focus on energy-efficient chillers, and an aggressive building-level energy reduction program known as the Energy Performance Improvement (EPI) program.

Kyle Field was part of the EPI and was an example of how sports venues can influence GHG reductions. Operational changes to the facility generated over USD 547,000 in one-time savings during the pilot period. Some building adjustments made include (1) the optimization of outside air operation for air-handling units and humidity sequences during winter, (2) better control of nighttime temperatures and events through discharge air temperature/static reset for units, (3) installation of backdraft dampers to minimize the intake of excessive outside air, (4) repairs made to non-operational valves and failed

sensors, and (5) reprogramming for better temperature control. Table 1 demonstrates the reduction in consumption and dollars saved through this initiative. A new baseline should be calibrated regularly to encourage good stewardship. Building systems require constant monitoring. Otherwise, the behaviors that led to increased (unnecessary) costs often resurface.

**Table 1.** The reduction in consumption and dollars saved through this initiative (Source: TAMU Energy Performance Improvement Website).

| Commodity | Projected Consumption | Actual Consumption | Avoided Consumption | Avoided Cost * |
|---|---|---|---|---|
| Electricity (kWh) | 14,691,450 | 13,377,293 | 1,314,157 | USD 102,504 |
| Chilled Water (mmBtu) | 68,441 | 45,962 | 22,479 | USD 350,854 |
| Heating Hot Water (mmBtu) | 13,427 | 7818 | 5609 | USD 93,682 |
| Total Savings | | | | USD 547,041 |

* April 2019–March 2020.

While lowering energy consumption and improving efficiency are vital and substantive contributors to GHG reductions, more can be done in event spaces to broaden the impact of these efforts. Communicating with the public about the need for and results of important GHG reduction work is vital in raising awareness. While TAMU has many growth opportunities to make events more sustainable, an initial step in raising attendee awareness about the sustainability of sports initiatives appeared on digital screen boards for the first time during the 2022 football season (Figure 2). An accompanying announcement expanded on the program.

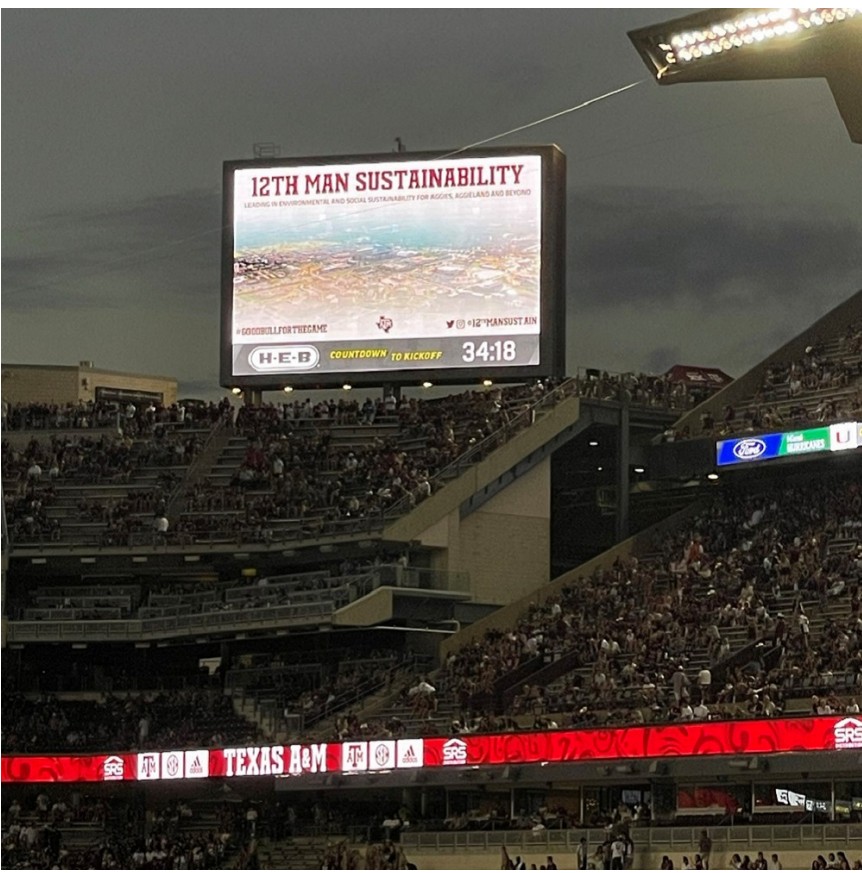

**Figure 2.** Kyle Field Screen Board: 17 September 2022, Texas A&M v. University of Miami at Kyle Field, College Station, Texas, USA (Source: K. Wellman).

*4.3. The TAMU Stars Report*

TAMU contracts with Gordian Inc. to regularly perform a GHG inventory for Scope 1, Scope 2, and Scope 3 emissions. Gordian Inc. uses SIMAP for emissions calculations. The most recent report generated in October 2022 highlights Scope 2 emissions, which, in TAMU's case, are indirect GHG emissions associated with the purchase of electricity, as the largest contributor to GHG (Scope 1: 116,082 MTCO2e; Scope 2: 201,022 MTCO2e; Scope 3: 45,982 MTCO2e).

The air emissions inventory was completed using the following methodology. The total natural gas consumed on campus was 1,980,896 mmBTU. Of this, 1,689,318 mmBTU was burned in units that have a Continuous Emission Monitoring System that calculates NOx, SOx, CO, and PM. The remainder was calculated using the "Potential To Emit Calculator for Boilers and Emergency Engines" published by the Environmental Protection Agency (EPA).

STARS (The Sustainability Tracking, Assessment & Rating System) in the United States is a transparent, self-reporting framework for colleges and universities to measure their sustainability performance (see https://stars.aashe.org/about-stars/, accessed on 30 November 2022). In its participant report, five categories (Academics, Engagement, Operations, Planning & Administration, and Innovation & Leadership) are evaluated for tracking a university's sustainability (see the example at https://reports.aashe.org/institutions/middlebury-college-vt/report/2022-03-04, accessed on 17 November 2022). Greening events is an important category here.

TAMU has participated in STARS since 2012 and has made annual submissions since 2015. It has served as the baseline by which the campus understands holistic sustainability. In fact, the SMP evergreen goals and targets were developed using a STARS gap analysis. For the first time in 10 years, the 2022 submission will include credit information for IN-18: Green Event Certification. This will reflect the Sustainable Event Certification program that was launched at the beginning of the year.

*4.4. TAMU Sustainable Event Certification (SEC)*

4.4.1. Initiating the Sustainable Event Certification Process

In April of 2021, it was decided that monthly newsletters would be sent out to everyone in the Aggie Sustainability Alliance (ASA), a campus-wide program that encourages students, faculty, and staff to participate in fostering a culture of sustainability. In these newsletters, the Office of Sustainability highlighted various environmental, economic, or socially sustainable events around campus. Looking at these sustainable events every month and noticing an increase in presentations at sustainability conferences about event planning, it was decided that Texas A&M should have its own version.

With the number of events happening all over campus, many environmental, economic, and social impacts could be minimized or mitigated with proper planning. Reviewing other schools across the nation, many had some form of sustainable event certification, and it confirmed that our university needed one, too. In the US, various academic institutions, such as Stanford University, Columbia College, Champlain College, and the University of Florida, have taken up the task of greening events as sustainable learning destinations. In addition to campus clubs and faculty departments being available to guide event organizers' faculty departments, they also provide checklists and certifications to encourage sustainability in events (see, for instance, https://sustainable.stanford.edu/take-action/events/plan-green-event, accessed on 15 February 2023).

4.4.2. Developing the Checklist

The first development phase was to look at a typical event planning process and determine areas of sustainable improvement. Most people think of the Waste Minimization checklist items when they consider sustainable choices. Reminders to have recycling bins available, use digital and social promotion instead of paper to advertise the event, and source reusable catering items, such as dining ware, utensils, and tablecloths, started the

list. From there, purchasing from local vendors and vendors who prioritize healthy staffing and environmental practices were the following checklist items added. Since many events hosted on campus are conducted by student organizations with limited financial support, encouraging borrowing items, writing digital thank you notes, and using reusable décor both encouraged sustainable behavior and assisted with financial constraints. Considering that $CO_2$ emissions from vehicle transportation account for a large negative impact on the environment, the checklist suggested hosting events on campus or virtually or carpooling.

Next, the creators of the checklist looked at The Campus Sustainability Hub through the Association for the Advancement of Sustainability in Higher Education (AASHE) to ascertain ideas and resources from other institutions. From this research, new checklist items were added, such as hotel accommodation guidance, excluding individually wrapped items such as sugar, salt, and coffee stirrers, and including an Indigenous peoples' land acknowledgment at the start of the event. This process continued until eight theme areas emerged, each with a space at the end to describe items selected in each section. Those categories and corresponding descriptions can be found below:

- Planning;
- Promotion;
- Food;
- Purchasing;
- Waste Reduction;
- Transportation and Location;
- Social Sustainability;
- Innovation and Bonus.

### 4.4.3. Planning

The items listed in the planning section encompass actions in the initial phases of event creation, such as receiving RSVPs, asking for dietary restrictions, determining whether the event will be live or virtual and recorded, establishing the event's purpose, and setting goals for the event.

### 4.4.4. Promotion

Promotion includes any action taken to market the event. The planner can review whether the event will be promoted digitally and accessibly through social media, bulk emails, and campus TV signage. If paper handouts will be used, looking at how the impact of waste can be minimized, such as using recycled paper and/or printing multiple handouts on one sheet, are other checklist items. The checklist encourages using reusable promotion forms, such as sandwich boards, yard signs, flags, or bus ads, that do not have date-specific information and can be used for future events. All items are listed to reduce waste.

### 4.4.5. Food

The selection and service of food and beverage for events have the potential for significant sustainability impacts, positive or negative. Offering vegetarian and vegan entree choices is environmentally friendly and includes dietary restrictions. Meat and animal products produce more negative environmental impacts, beef more so than chicken. Food service from a buffet minimizes the amount of packaging (usually plastic or Styrofoam) ending up in a landfill. It lets attendees choose what goes on their plates, which can minimize food waste. Though there is the additional cost of staffing a served buffet, often referred to as cafeteria-style service, such service limits portions and choices and can limit food waste even more than traditional buffet service. Texas A&M is currently the largest fair-trade University in the nation, and this checklist encourages purchasing fair-trade products. This movement changes the way that trade currently works. It aims to have better prices and better working conditions for employees and enables workers and farmers to have more control over their futures. Another way to incorporate sustainability into

food at events is to introduce attendees to foods from other cultures, which is a great way to promote social sustainability and have people try new things. Due to limited transport and fertilizer usage, serving local and in-season produce has a much smaller environmental impact than imported or out-of-season produce.

### 4.4.6. Purchasing

The value of a distinct checklist category related to purchasing is two-fold. First, many events on our campus are hosted by student organizations with limited funding sources, so finding ways to save money is very economically sustainable. Second, supporting local and sustainable companies is a great way to exercise both environmental and social responsibility, limiting transportation emissions and providing income to local community members. When venues do not provide in-house custodial services, sometimes event managers need to purchase their own cleaning products. Investing in earth-friendly, biodegradable products helps the planet and also the individual event's budget because the items are intended to last longer than disposable products.

Crafting tablescapes and centerpieces from natural items can be an inexpensive way to create a welcoming look at banquets. Additionally, instead of disposing of centerpieces after one use, creating ones that can be reused or provided as attendee gifts or amenities keeps those items out of landfills. Locally sourced and in-season florals and plants fund the local economy and limit transportation emissions and fertilizer use. Renting potted plants to create ambiance at events such as award ceremonies allows the plants to continue to be in use for other events in the community without the cost of permanent purchase. Table and chair rentals support local businesses and limit the cost of purchase and the need for storage space.

Many events on this campus have t-shirts or other promotional items as takeaway items for guests: utilizing sustainably made items and purchasing from Historically Underutilized Businesses, local businesses, or minority-owned businesses are great ways to invest dollars responsibly. After an event, it is customary to share appreciation with donors, volunteers, and other key stakeholders for the event's success. Electronic thank you notes may provide all appreciation needed, yet sending a sustainable or consumable gift provides a way to not create waste.

### 4.4.7. Waste Reduction

Waste reduction is most commonly associated with sustainable choices. Ensuring that more than just landfill bins are provided at an event can drastically reduce the waste heading to a landfill, as many items are now made from recyclable materials. Proper signage is also important, so attendees know their waste options and what can go in different bins. Eliminating disposable items or finding reusable alternatives for production items, such as tablecloths, napkins, drink dispensers, dining ware, condiments, name badges, and coffee accessories, can significantly reduce waste. Workforce areas can also have additional separation for compostable materials, as training on proper disposal can be more readily provided to these populations.

### 4.4.8. Transportation and Location

Transportation emissions account for a large amount of GHGs worldwide, so minimizing them is a significant way to make an event more sustainable. This can be as easy as hosting it virtually or having a virtual option to decrease the number of people attending in person. In addition, holding events outside or on campus is an easy way to do this, as many event attendees are already on campus daily and do not have to make an additional trip to attend an event. Holding events outside or in sustainable buildings can help reduce energy costs for the building. If an event is held off campus, the checklist asks meeting planners to encourage guests to carpool or utilize alternative transportation, such as biking or using a city bus, to decrease the number of single-occupancy vehicles traveling to the same location. If an event is overnight and hotels are being used for attendees, choosing one that is in a

central location to the rest of the event can limit travel, especially for attendees who did not drive themselves. Along with that, encouraging attendees to still implement sustainable behaviors in their hotel, such as unplugging electronics when not in use, adjusting the room temperature while gone, and bringing their own toiletries, can improve several areas of sustainability.

### 4.4.9. Social Sustainability

Social sustainability blends traditional social policy areas such as equity, diversity, and inclusion with social issues such as justice, economic opportunity, participation and influence, community and global needs, and well-being and quality of life. This is a critical element of sustainability that is often overlooked; therefore, in an effort to produce more visibility through the checklist, the creators felt it was important to designate it as a distinct section. Encouraging attendees to support local non-profits or community organizations through financial or item donations supports local sustainability. Land acknowledgments are a way to acknowledge the Indigenous people where an event is being hosted. This affirms continuous Indigenous presence and rights, acknowledges the ongoing effects of settler colonization, and supports Indigenous people's political, legal, and cultural sovereignty. Ensuring that the event is accessible to people with disabilities, whether in person or virtual, provides all attendees with equal access to event content. In the USA, the Americans with Disabilities Act (ADA) provides guidance and standards for event and venue managers to use. Communications for the event should be available in languages commonly used by guests to provide a more inclusive environment. Partnering with campus or community organizations that champion social sustainability work, such as LGBTQ+ rights, women's rights, racial justice, etc., can not only support their causes but also provide additional access to interested parties when they assist with event promotion.

### 4.4.10. Innovation and Bonus

This section was added because the office knows this list is not all-inclusive and there are many more ways to incorporate sustainability into event planning. Planners are encouraged to be creative and find other ways to earn double points if approved by the Office of Sustainability. With the basic framework of the checklist built, the next step was to determine what program would be used to create the certification. Options such as Microsoft Excel and Canva were considered, but it was determined that Adobe InDesign would be best due to the Interactive PDF feature. This allowed fillable text boxes, drop-down menus, and checkboxes to be incorporated. Creating the document took longer than expected, as using InDesign was a skill everyone in the office needed to improve. Several versions later, the checklist was ready for testing and feedback.

### 4.4.11. Tiers and Tie to ASA

From the beginning, it was known that a tiered system would be the method to score the events. Though the certification was separate from the current ASA checklists, the decision was made to tie the tiers back to the ASA. Events could certify at either the Supporter (30% of checklist items were met), Advocate (55% were met), or Champion (80% were met) tier. This was to provide consistency among the different certification programs offered by the Office of Sustainability and an incentive for event planners to make more sustainability impacts to reach a higher tier.

### 4.4.12. Icon

Though the Office of Sustainability wanted the SEC to be tied to the ASA in terms of tiers, it was the hope that this checklist would stand as its program with its icon. Because event planning happens across campus, the icon's design needed to be TAMU-centered. It was decided to create a skyline of the major buildings across campus. Several versions were hand-drawn using a circle with a banner created on Canva, a graphic design website as illustrated in Figure 3.

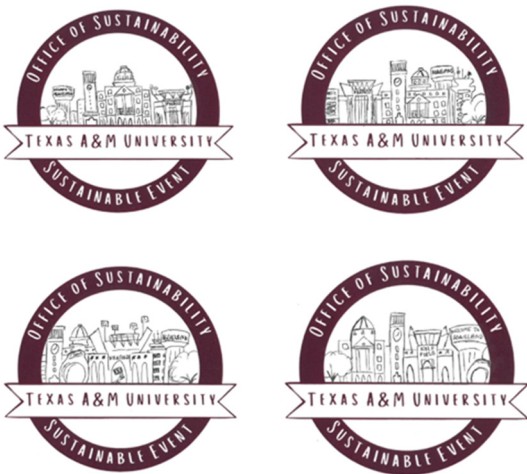

**Figure 3.** Original hand-drawn sketches of potential SEC icon.

Option 2 (top right icon in Figure 3) was selected, as it included a wider variety of campus landmarks but did not have one building at the center focal point. With that decided, it was time to create it digitally. As the original person who volunteered to create the icon was no longer available, the office had to wait until a new Graphic Designer started in January 2022 for it to be completed. The final icon for the SEC can be seen in Figure 4 below.

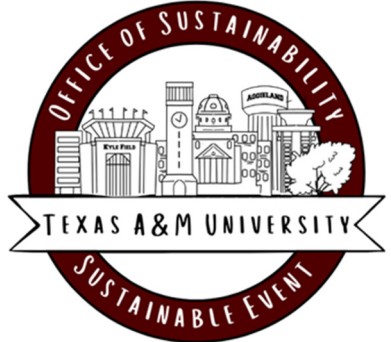

**Figure 4.** Current SEC icon.

In the ASA, each tier that someone certifies in is tied to a specific color. To maintain consistency, the icon was also created in each of the colors corresponding to Supporter, Advocate, and Champion shown in Figure 5. When an event certifies, they have the option to use the maroon icon or the appropriate tier color on any of their promotional items to let others know that their event is a certified sustainable event.

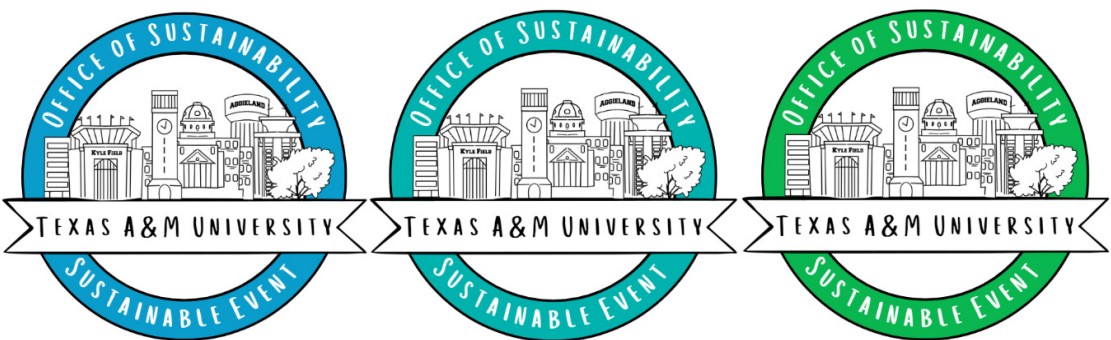

**Figure 5.** Current SEC icons based on tier certified (Supporter, Advocate, and Champion).

### 4.4.13. Resources

To simplify the event certification process and encourage event planners, different resources were developed to aid their event planning.

Most events and many checklist items center around food. A list of all Bryan/College Station restaurants was generated. Well-known fast-food chains were immediately removed to shorten the amount of research needing to be completed. The history of each of the restaurants on the shortened list was researched. The office looked for anything that alluded to the restaurant being started in the area, being Aggie owned, being locally owned, etc. If it could not be determined whether the restaurant was local, it was removed. Some restaurants are not technically local but focus on sustainability in their core values, such as *Snooze, an A.M. Eatery*, so they were kept in the resource. People are unlikely to look at a large spreadsheet or document of restaurants, so effort was dedicated to creating a branded document in Canva that was easy to navigate (see Appendix A).

Once food options were secured, the following resources created were for rentals and florists. As there are fewer of these companies in the area, this resource was quick to complete. Like the restaurants, research was conducted into the history of each, and it was determined whether it was local/sustainable or not. Similar branded documents were created for each (see Appendices B and C).

One of the checklist items is for events to be hosted in sustainable buildings on campus, but that information is not easily accessible. It seemed obvious to look at Leadership in Energy and Environmental Design (LEED)-certified buildings, but only a few are on campus. Multiple buildings on campus are built to LEED standards but are not certified. It was decided that these would still be considered sustainable buildings. Appendix D has the official sustainable building resource.

One aspect that will constantly be evolving with this certification is the resources. The office aims to be as transparent as possible to those considering submitting their event. Providing resources is an easy way to minimize the amount of work someone has to do. The downside is that those restaurants close while others open, buildings could be updated or new ones built, and rental companies and florists could go out of business while others take their spot. These resources have already had to be updated once, and significant changes were made, as COVID-19 is still impacting businesses, while economic development is blooming in other parts of the community.

### 4.4.14. Branding/Event Signage

One crucial aspect of program development is brand and brand recognition. As mentioned previously, the tier colors were tied to ASA tier colors. The office wanted to tie in other brands around campus, specifically with event signage and Utilities and Energy Services (UES) Recycling Services. The recycling and composting event signage colors are pulled from the campus standard recycling branding and illustrated in Figure 6. The event signage can be found in Appendix E.

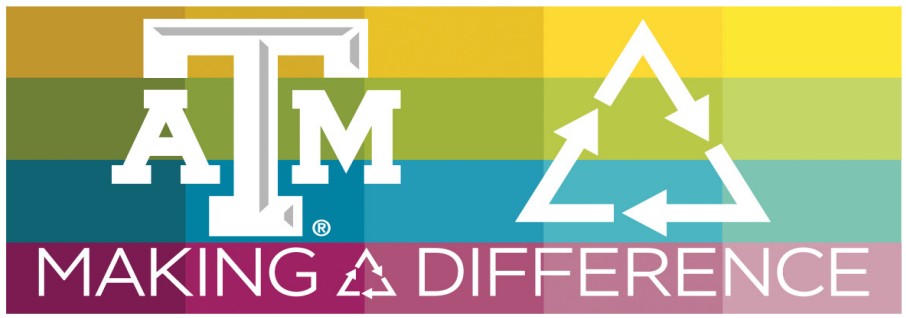

**Figure 6.** Sample of Utilities and Energy Services Recycling signage.

#### 4.4.15. Incentives

The following steps for the checklist were to decide on incentives and how to promote the proper certification promotion. Because of the following and social media interaction that OS has, it was determined that one of the significant incentives for certifying an event was that OS would be a major partner in marketing and promoting the sustainable event. The word would be spread to thousands of campus members through the ASA newsletters, promotion on the OS website, and on OS social media channels.

#### 4.4.16. Promotion

With the checklist in good shape, resources created, and incentives decided, OS was in a position to launch the program. As April was approaching and OS was hosting a month-long program for Earth Month, the Annual Sustainability Awards Breakfast was the perfect first event to be certified and to announce the official SEC Checklist (Appendix F) launch, as it was the first ceremony back in person and was being live-streamed. This event could be nothing short of Champion Level, so an initial audit of the event was conducted to see where the event was percentage-wise without any additional efforts. Most of the event checklist items are already norms for OS, so the event was very close to Champion. Efforts such as including a land acknowledgment, making the centerpieces not only sustainable and a giveaway item but also completely edible, and utilizing a company that focuses on sustainable award production bumped the percentage to 81.9%, breaking the Champion threshold.

Toward the end of the event, it was announced that everyone in attendance and watching live was witnessing the first Texas A&M University Certified Sustainable Event. Post-event, OS began posting on social media and through the ASA to promote the new event. Unfortunately, this was near the end of the semester, and summer break decreased the number of events happening around campus. Therefore, there were few events left to keep any momentum going. Similar efforts and the addition of bulk emails were used for the Fall 2022 semester.

#### 4.5. Implementation and Testing

Once the initial draft was created, the checklist was sent to the events classes for testing. The Office of Sustainability received the following three test events: The Hunt at Century Square, Tee Off for REACH Golf tournament, and The End of Year RPTS Recognition Banquet.

#### 4.5.1. The Hunt at Century Square

This event was submitted at 35%, but several of the items listed in Innovation and Bonus really fell under existing checklist items, so they did not count. For example, the following is the feedback the event planner received:

- "This event will be held using a virtual platform to eliminate any waste" was listed in the bonus section but would fall under the first point in Transportation and Location— "The event will be held completely virtually or have a virtual option."
- "Participants will be encouraged to ride the bus" was listed in the bonus section but would fall under "Carpooling and use of alternative transportation will be encouraged."
- "Participant gifts will include reusable mugs" was listed in the bonus section but would fall under Food and Purchasing's "Any thank you gifts will be sustainable or consumable."
- "We will feature tasks that encourage sustainability" was listed in the bonus section but would fall under Social Sustainability's "The event will be focused on environmental, multicultural, or equity-related topics."

This submission showed immediately that there needed to be a description of the event on the first page, as there needed to be a way to know what the event was otherwise. The final total for this event was 32.2%.

### 4.5.2. Tee off for REACH Golf Tournament

This event was submitted at 66% with only one item of feedback:

- "It was clicked that all promotion for the event would be done digitally or paperless but three of the paper promotion items were checked. The paper promotion items were unchecked making the new total for that Section 5."

The final total for this event was 64.5%.

### 4.5.3. The RPTS Recognition Banquet

This event was submitted with 26 (41.9%) checklist items marked, but the planner listed that were 36 (58%). The following is the feedback that was given to the planner:

- "The item "Event will be held on campus to minimize transportation emissions" was checked but the location on the first page says it will be held at the *Ice House* on Main."

This submission showed that a way to auto-calculate check boxes would be very beneficial as the planner may not always do the correct math. Unfortunately, Adobe InDesign does not offer that feature in the Interactive PDF creation as of now. The event's final total was 25/62 (40%).

### 4.5.4. Feedback

Several different people and classes provided feedback for the initial version of the SEC. First was an Instructional Assistant Professor who taught event management and operations, examined the draft framework, and offered feedback:

- "One of the things I did point out was the importance of balance. While I know you would like the events to do as many of these things as possible, some are mutually exclusive within a category and others, in combination with other choices, make for traditionally poor event outcomes. So, while I wholeheartedly agree with what you're doing, I do want to caution you from setting the bar too high, or allowing the "points" to be cumulative, rather than only a certain number per group, so that evidence-based practices in event management aren't violated in extreme efforts to reach sustainability goals."
- "Without actually seeing it applied to an event, my first thought is that your percentages are likely good, though 80% seems quite high, based on what I stated above, but I suppose it could be possible. Could we try those out with my Fall students' events? Then we can see where those percentages fall and if it makes sense? I don't want to make the process too complicated for you, but I think the percentages should somehow reflect multiple categories in the form."

Other checklists and certifications created by OS have shown that people around campus are hesitant to take on additional tasks that are over-complicated, so the decision to keep the checklist as one percentage calculation versus percentage calculations for each section was made.

Next, teams of students in two event management classes in 2021 and 2022 examined and implemented the draft framework. Lastly, the OS Outreach and Education Intern student reviewed the checklist and provided feedback regarding the layout of the checklist having more spacing between the title and checkboxes and between each section, ensuring that points did not counter each other, that accessibility was considered, and that the tiers were clearly labeled. All of these changes were taken into consideration and adjusted appropriately for the final version, which launched in April 2022.

### 4.6. Implementing the SEC Draft in Event Management Courses

The examples below illustrate how event management students at the Research 1 institution in Texas, USA, have begun to assist local event organizers (their clients) in contemplating and implementing micro-level sustainability measures. Funds provided by a small, internal university grant incentivized the students to implement aspects of the new SEC at the onsite events they planned and implemented for the campus or community client.

As noted above, the SEC addresses environmental and social criteria for sustainability, mindful of the interrelatedness of social and environmental inequity and inequality in events [53]. Planning and policy items are also included, attentive to addressing social as well as environmental aspects. Ballet et al. (2020) emphasize the importance of including policy dimensions that address factors such as social cohesion, equity, and safety [54].

Societal health and safety must be a more significant factor in future SECs. However, climate change and overall environmental sustainability are existential threats, and advancing student learning, skills, and leadership to address these in the event management field is a critical priority. As the earlier discussions in this paper show, sustainability and destination resilience are beginning to take greater priority with increasing awareness of the importance of disaster planning and adaptive management to handle extreme weather and threats such as pandemics (COVID-19 was a wake-up call in this regard). The examples below show that student efforts on campus and the local community collaboratively initiated small changes in practice, fostering continuity in sustainability actions, engaged learning, and critical thinking, as students often grappled with trade-offs in the sustainability desired versus organizers' needs to enable the event to proceed.

We will discuss below a couple of examples of the students' use of grant funding provided over a three-semester period in 2021–2022 to facilitate sustainability in local events with the help of the draft SEC that was being pilot-tested. The small financial incentives (USD 500 per team per event) led to creative thinking and alternative designs the student teams developed or purchased from the funds, while they grappled with the trade-offs that often needed to be made. Cooperative learning and collaborative knowledge sharing were fundamental principles as students engaged with event clients and the event instructor, as well as vendors and suppliers. The instructor also interacted with and provided feedback to the OS SEC planner for continuous learning and the adaptation of the SEC Checklist throughout the draft testing process.

### 4.6.1. An Outdoor Event Serving Texas Barbeque

In Fall 2021, students implementing previously planned events were asked to evaluate their events against the new SEC, but they were not required to change their plans or designs, as implementation technically began in the Spring 2022 follow-up course. However, one group, planning an outdoor fundraising banquet, requested permission from their client to change the plastic silverware to more sustainable biodegradable bamboo and proceeded to implement this. At first, the guests at the banquet thought the material to be quite odd, but they were quickly impressed by its sturdiness and were grateful to know they were a part of that sustainable event decision. Food was also locally sourced and staff members were also trained locally, contributing to social sustainability. Offering opportunities to donate funds in support of a non-profit was an additional decision supporting this event. Texas barbeque was requested by the client. Though our students were aware of this negative impact, they maintained the beef dinner in support of their client's desire for Texas barbeque catering. As discussed during the initial draft creation of the SEC Checklist, an event team could inevitably be faced with making such trade-offs on sustainability in order to support their client's needs or vision. This was one such case.

### 4.6.2. Tailgate Event at College Football Game

American football weekends can be filled with many ancillary events. One is a "tailgate", or an outdoor casual picnic that can be catered and used as a way to gather with friends and fans of a sports team before the match. The event management students in the Fall of 2021 planned one such tailgate. Rather than purchasing new decorations, the students investigated several storage closets of the client, finding historical items and branding merchandise that could be reused for décor and takeaway items at the event. They purchased pre-cycled and biodegradable eating containers and takeaway boxes for food. Additionally, they served beverages in large serving containers to not add to waste. They selected in-season fruit to serve as a side dish. They chose a local food vendor and selected

chicken on the menu to demonstrate their understanding and commitment to sustainability. However, due to a misunderstanding over the number of guests in attendance, there was a significant amount of leftover food. While many students and guests could take home some leftovers and the department ate well for several days after the event, this miscalculation resulted in large food waste. While some steps were taken with great care and intentionality to support sustainability, others were missteps. Students learned to take responsibility during these event implementation processes by being empowered to make decisions and to learn from them, both from the positive outcomes and the errors.

4.6.3. Building on 2021 Sustainability Practices

As the Spring of 2022 arrived, several student event groups used sustainable event funding to purchase higher-cost and higher-quality items that support the environment physically and socially. In the Fall of 2022, the organizers of a few recurring events that in the previous year had made some sustainable choices returned to those practices, and some new events explored new choices.

In Fall 2022, a Family Western Night with dinner and dancing for nursery school children, their parents, and teachers changed their centerpieces in support of sustainability. They had planned to arrange cut flowers and discard them after the event. However, when offered sustainable event funding, they found a local plant nursery, where they purchased in-season potted plants to serve as the focal point of their centerpieces. Their client saved large aluminum cans in their kitchen over several weeks. The students placed the pots inside and then wrapped them with swatches of fabric the client kept on hand for craft projects (Figure 7). The result was a creative display that reused materials and provided a lasting gift, as they were then presented to the teachers at the end of the evening in appreciation for their service.

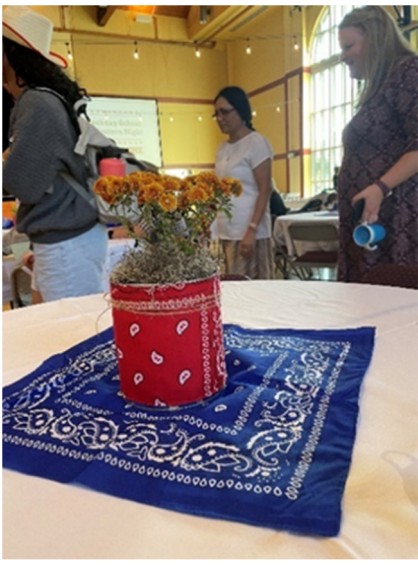

**Figure 7.** Sustainable centerpiece and gift using reused materials and locally sourced plant (Source: D. L. Sullins).

Another event that changed direction when supported by financial incentives was a local multi-use facility that hosted a community night for costumed children to receive candy in celebration of Halloween. One of the activities at the event was pumpkin painting. Rather than using traditional paints, the students purchased all-natural paint powder to mix with water at the event. The vendor selected was also woman-owned and used wind and solar power to create its products (Figure 8). Since the paint functioned differently than they were accustomed to, it required some trial and error to find the perfect consistency to create the desired colors.

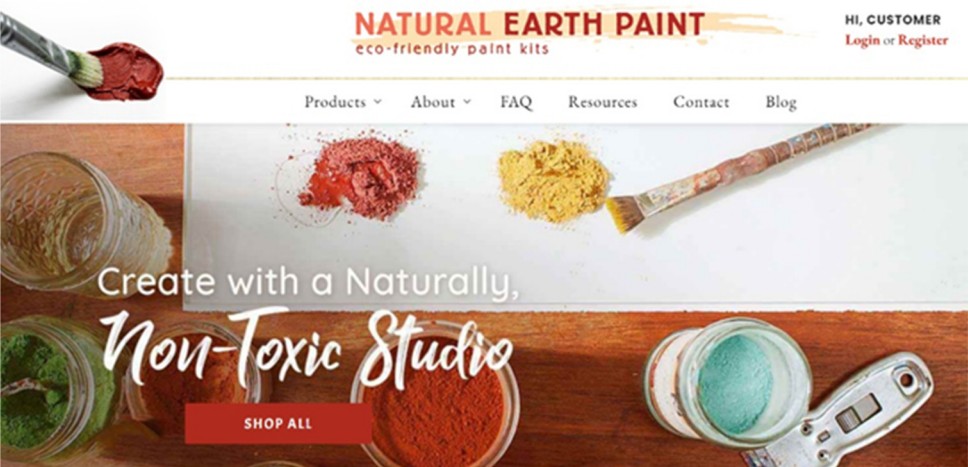

**Figure 8.** The all-natural paint powder the vendor selected (Source: https://naturalearthpaint.com/, accessed on 5 January 2023).

At the Fall 2022 tailgate, event students decided to craft high-quality reusable centerpiece flower boxes (Figure 9). The boxes themselves were made of recycled cardboard, but the flower product was made of foam. Once again, although sustainable choices exist in the green event management plan, event management students selected other choices intentionally, not in support of sustainable event practices, in order to conduct a well-executed event that both met the event organizer's goals and showed respect for sustainability.

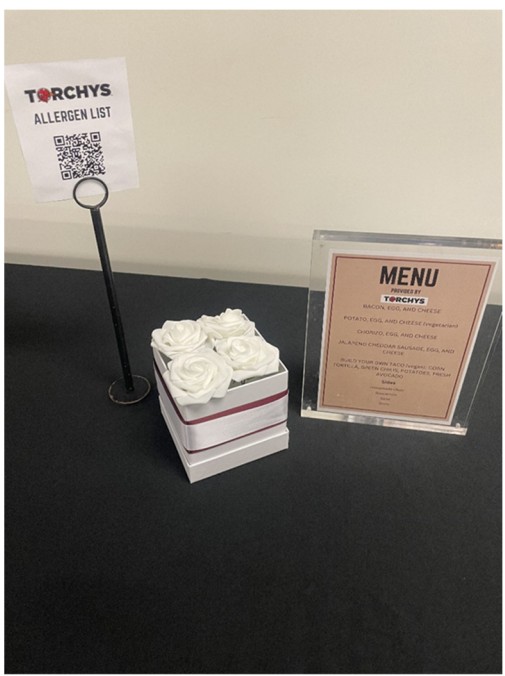

**Figure 9.** Flower boxes made of recycled cardboard and planned for reuse (Source: D.L. Sullins).

Again, the outdoor banquet organizers used bamboo utensils for the event. However, they did not request funding for this iteration of their event, seemingly embodying a new practice in their event plan that they now saw as their own. Again, these are encouraging signs that event clients will come to see what more is possible, get in the habit of making those changes, and then embrace them as commonly accepted practices in their event plan, rather than being a one-off decision agreed to by the client just to receive additional funds.

### 4.7. Implementing the SEC at Other Campus Events

Annual Sustainability Awards Breakfast

The first event to officially be certified through the SEC was the annual Office of Sustainability Awards Breakfast. This is traditionally hosted in person to honor those who have supported the Office of Sustainability and announce the Sustainability Champions and STARS award winners. The Sustainability Champion Awards aims to recognize and reward individuals—students, faculty, staff, and team members—who have demonstrated exemplary effort, dedication, and leadership throughout the year in fostering a campus culture of sustainability. The completed checklist for this event can be found in Appendix G.

The planning for this event started in January 2022 and was hosted on 22 April 2022. An invitation list was sent out to individuals whom OS wanted to attend in person in March so that RSVPs and a headcount for food could be determined. The Save the Dates were sent electronically to reduce paper waste. The event was live-streamed to include as many people in the celebration as possible. Bulk emails, social media, and TV ads in various buildings across campus were used to promote the live stream. The only paper present at the event were quarter-sheets of codewords for the month-long giveaway, which were printed on recycled paper.

The office coordinated with Aggie Dining to provide a buffet-style meal with fresh, seasonal fruit, Rosenthal sausage (made on campus), overnight oats with berries, local jams, eggs, French toast casserole, and a fair-trade coffee service.

For setting the tables, reusable tablecloths, napkins, tables, and chairs were all rented from the MSC. No individually wrapped or single-use items were provided by Aggie Dining. The office decided to partner with TAMU Urban Howdy Farm to create edible centerpieces. The office went to the farm the day before the event and harvested fresh lettuce, spinach, herbs, and edible flowers to create the centerpieces. At the event, one chair at each table had a sticker on it, and the person sitting in that seat was able to take the centerpiece home. The centerpieces being edible was awarded an Innovation and Bonus point. All other décor at the event was from previous award ceremonies and events.

The awards for the Sustainability Champions and STARS winners were purchased from EcoPromotions. This is a company that specializes in sustainability for promotional items, as the awards are made from bamboo and recycled glass. Purchasing from this company awarded the event an Innovation and Bonus point.

The event focused on celebrating the accomplishments of the award winners, but it also focused on environmental and equity-related topics impacting the campus. The event was opened with a land acknowledgment as a reminder of the Indigenous land that Texas A&M University sits on, so several items under the Social Sustainability section were points earned.

Most of the event checklist items are already norms for OS, so the event was very close to Champion to begin with, and very little additional effort was needed to cross that threshold into the next tier. With proper planning and the checklist in mind, this was the perfect event to launch the SEC, as it was a traditional event with sit-down food service. It was realized a few months later that the SEC was not as applicable to other types of events.

Most recently, the Office of Sustainability certified their annual Campus Sustainability Day through the SEC (see Appendix H). This event was revealing for the certification, as it was not a traditional event with food being served. Because of that, a number of checklist items were not available, and others, while technically true, were a stretch. For example, fair-trade products were handed out from Aggie Dining, so that point was achievable, and because only snacks were handed out, meat options were technically minimized. That point was counted, but it was done so through a technicality.

Because this was a tabling event for sustainability education, dietary needs, center-pieces, and reusable dining items (pitchers, dining ware, etc.) were all not applicable, meaning several points were missed out on just because it was not a sit-down-to-eat event. This information is currently being used to update the SEC to be more inclusive of non-traditional events.

*4.8. Submitting Event Certification to the TAMU STARS Report*

In the 2022 TAMU STARS Report, Innovation Credit 18—Green Event Certification was submitted for the first time (Figure 10). Though this submission has not been reviewed by AASHE at this time, it is expected that full points (0.50/0.50) will be awarded.

**Green Event Certification**

| Status | Last Updated | Possible Points | Points Earned |
|---|---|---|---|
| Complete | 12/02/22 2:49PM | 0.50 | 0.50 |

\*Required Fields \* Conditionally Required Fields
**Reporting Fields**

\*Does the institution have or participate in a green event certification program?

| \* Yes ⌄ |
|---|

*If yes:*

\*Has the institution held one or more certified events in the previous year?

| \* Yes ⌄ |
|---|

Does the institution's green event certification program address the following?

| | Yes or No |
|---|---|
| Sustainable transportation options, teleconferencing options, and/or carbon offsets | \* Yes ⌄ |
| Sustainable catering (e.g. sourcing local and third-party certified food and beverages, providing vegetarian/vegan options, using reusable/compostable materials) | \* Yes ⌄ |
| Paper consumption (e.g. minimization and recycled/FSC certified content) | \* Yes ⌄ |
| Energy efficiency (e.g. equipment and lighting) | \* Yes ⌄ |
| Waste minimization and diversion | \* Yes ⌄ |
| Communications and/or signage about the sustainable practices | \* Yes ⌄ |

**Figure 10.** A 2022 STARS IN-18 Green Event Certification Submission.

## 5. Discussion

Event sustainability on the main TAMU campus is situated within the context of destination resilience and the learning destination. However, our approach emphasizes that the learning destination is a complex destination system. Similar to the learning organization that focuses on collaboratively creating and using diverse knowledge within the organization [37], cited in [10] (p. 347), the learning destination draws on practical knowledge and developing soft skills to adapt collaboratively and swiftly to emergent wicked problems, as well as practical challenges that arise in striving toward sustainability in complex destination systems, including the diverse interests of multiple stakeholders in the destination.

As shown in the case above, numerous initiatives have been undertaken to reduce Scope 1, 2, and 3 emissions and strive toward being a carbon-neutral destination. Within this larger context of destination resilience arose the objective to green events and develop and test a sustainable event framework that included certification and a checklist. Communicating sustainability-oriented events to faculty, staff, and students across campus via the Aggie Sustainability Alliance (ASA), knowledge sharing at a global conference as well as exploring efforts by other academic institutions contributed to jumpstarting this step. Determining criteria to evaluate and their weightings in the draft SEC that was designed, collaborative user testing by students in event management classes as well as implementation at other events, and receiving joint feedback and implementation actions from event hosts all led to revising and tweaking the draft SEC toward the final version launched in 2022. Adaptive planning, inclusivity, and the participation of these key stakeholders were demonstrated in the overall process, which comports well with the approach outlined by Sadd et al. (2017) [10]. While the overall events in the learning destination's portfolio

vary from year to year, some events recur annually. The SEC was grounded in the local destination context, and continuous learning occurred over time, as the case illustrates.

The SEC was created to encourage event planners across campus to implement more sustainable practices in their events and further facilitate destination resilience. Providing them with education, a framework, and resources can help so that people do not have to start from scratch. The SEC facilitates destination resilience through sustainability practices, collaborative learning, and joint participation in the circular economy. It focuses on collaborative engagement between the students (volunteers) and local vendors and service providers, as well as local resources, including locally sourced foods (hence contributing to food security as well). Soft skills and practical knowledge were gained and applied during pilot testing. Students grappled with sustainability trade-offs as they sought to implement good actions, such as reducing waste, reusing, and otherwise recycling as much as possible. They exercised empathy, seeing the event from the perspective of their clients and facilitating awareness, learning, and change toward greener practices by working collaboratively with them.

Examples of sustainability actions undertaken by the event management students and their collaboration with local vendors demonstrate engaged learning, knowledge gains (and transfers), and valuable soft skills being built in the learning destination. It also illustrates how students and local event organizers and businesses may potentially become eco-champions and stewards of green events, fostering the awareness raising of sustainability practices (e.g., reducing plastic use) and potential ways to reduce carbon emissions (e.g., facilitating greater access to public transport and campus buses using biodiesel, as well as e-bikes, which are currently successfully being used). The term championship refers to an individual who encourages or is an advocate of a particular cause or way of thinking, while stewardship denotes conservation actions facilitated by a belief in the seriousness of environmental problems [55]. Mair and Laing (2012) characterize the eco-champion as a catalyst for pro-environmental behavior [32].

The sustainability practitioner requires theoretical and practical knowledge and skills gained by interactive experience and engagement with environmental, economic, social, and cultural issues in the local community space and the wider regional/global tourism system. Direct participation at the community level provides the opportunity to develop core literacies, where literacies refer to "an integrated set of skills and knowledge that are practiced and refined reiteratively to facilitate culturally thoughtful and ethical practitioners" in the learning destination and event management domain [52] (p. 138). An important goal and effect of such participatory action is collaborative learning, which results in synergistic, joint learning, knowledge building, and skill acquisition [56], cited in [52] (p. 137), developing and exercising practical knowledge (*phronêsis* or practical wisdom in the Aristotelean sense).

Students should not only possess certain discrete and measurable areas of knowledge and skills but also be able to demonstrate practical and applied competence in dealing with diverse and complex problems. In pedagogic terms, different types of knowledge (for example, implicit, explicit, ritual, inert, tacit, and so on) become integrated when literacy is achieved [52] (p. 138).

The six literacies identified by the authors are shown in Table 2, along with the criteria that characterized learning destinations striving for resilience and sustainable event management, as well as examples from the TAMU learning destination case.

**Table 2.** Key literacies and criteria of learning destinations striving for resilience and sustainable event management, as well as examples from the TAMU case.

| Literacy | Description of Literacy * | Learning Destination Criteria | Example from TAMU Case |
|---|---|---|---|
| **Analytical literacy** | Skills, techniques, and qualities needed to undertake problem solving, issue identification, and critical inquiry, evaluate sustainability discourses and acts, etc. | Impact identification and green event management skills; event sustainability policies, frameworks and certification, etc., to help guide event development, management, and marketing in the complex destination system | SEC developed by brainstorming and analyzing information from other schools; Campus Sustainability Hub through the Association for the Advancement of Sustainability in Higher Education (AASHE) was examined to inform the SEC, etc., undergraduate courses in event management teaching necessary skills in classroom and through experiential education (direct participation in the event community) |
| **Ecological literacy** | Awareness of human–environmental relationships and consequences of decisions and actions that affect these. The connections extend beyond ecological to community, social–cultural, and built and natural environments. | Being especially aware of the interrelatedness, emergence, and long-term effects on ecological, social, and cultural sustainability and well-being; drawing on the circular economy, etc. | Numerous initiatives to reduce Scope 1, 2, and 3 emissions; encourage green transportation; recycle plastic to reduce environmental impacts (and impacts on human and ecological health); drawing on the circular economy for sustainable food choices, etc. |
| **Ethical literacy** | Development of values and ethics within students' thinking, and how these may be gained not just through classroom learning but also as practical wisdom (*phronêsis*), through direct experience and collaborative learning | Exercising practical knowledge (practical wisdom to make good decisions on sustainability trade-offs and deal with ambiguity and emergent issues in complex systems); creativity in designing solutions with empathy; attentiveness to disadvantage and inequity | Students grappled with ethical choices on how to advance sustainability actions when event client was disinclined to undertake the action; students exercised empathy and engaged in awareness raising and collaborative learning with clients. SEC attends to equitable access for people with disabilities. |
| **Multicultural literacy** | Appreciation of diverse knowledge (local, Indigenous/traditional, scientific, technical); appreciation of diverse cultures, values, interests, and voices, attentive to disadvantaged ethnic minority and marginalized groups. | Awareness raising and educational skills to inform and engage diverse visitors to contribute to destination resilience and sustainability of place and events | Diverse perspectives, genders, and types of knowledge were invited and welcomed to assist with campus-wide sustainability initiatives, including the SEC development efforts. |
| **Policy and political literacy** | Appreciation for the way green events are planned and managed, how decisions are made and how implementation occurs, the influence and power of stakeholders, collaboration, and issues of power, trust, and influence | Collaborative engagement among key stakeholders; proactive/adaptive planning and disaster preparedness (policies, training, strategies, etc.); collaborative learning | OS developed policies for TAMU sustainability; collaborative engagement between OS, students, and other campus stakeholders; event management students collaborated with event clients and vendors. |
| **Technical literacy** | Theories, concepts, and frameworks that provide technical knowledge (including disaster planning, crisis management); planning, management and marketing techniques, conservation tools and strategies, etc. | Technological skills to address online and onsite event marketing, etc. | OS staff exercised technical expertise to design various marketing and promotional materials, graphic designs, branding, signage, icons, websites, etc.; technological skills developed in classroom and in practice by event management students. |

* Description adapted from [52].

## 6. Future Actions for Destination Resilience and Greening Events

The current goals for 2023 are to have five events certified across campus. The OS also hopes to receive full points in IN-18 Green Event Certification in the 2022 STARS report. From here, OS hopes to implement a virtual sustainable event certification that focuses solely on virtual events, as these have increased since the pandemic. OS would also like to find a way to measure the impacts on sustainability from these events, for example, waste diverted instead of taken to a landfill, a reduction in GHG emissions, an increase in social sustainability education, etc.

Future factors to be considered for inclusion in the SEC are (i) providing QR codes for visitors to learn more about the sustainability efforts being undertaken at the event and how they can contribute; (ii) providing information on public health and safety related to recent airborne infectious diseases that may become endemic (the "new normal", as is expected of COVID-19); (iii) using virtual or augmented reality to provide experiences of the event to those unable to attend, including persons with disabilities, or those wishing to reduce their carbon footprint).

Hybrid conferences combining a live dimension for those attending in person with an online component for a virtual audience may also be an attractive consideration to offset the carbon cost of staging and traveling to events. Encouraging visitors to use carbon offset strategies is also a potential action step to add to the SEC in the learning destination, for example, by selling tickets through a combination of money and individual carbon credits [34].

The role of the public sector will also be increasingly important in policymaking for destination resilience and event sustainability during crises, e.g., to support small business owners in accessing capital for employee retention during future pandemics [57,58]. Future research is recommended to explore event organizers' abilities to cope with crisis scenarios without necessarily having to rely on state intervention, says Sigala (2020) [58]. "Developing strategies for a quicker and more effective conversion of presential to online events, setting up national or regional guarantee funds amongst events' organizers, or optimizing the adoption of tighter preventive sanitary measures during events should be among the future research topics in the context of COVID-19 and the events' industry" [59] (p. 423).

Additionally, Rowen (2020), cited in [59], suggests that future events will have to play an even more important role in consumer education and that lessons from COVID-19 could offer the industry opportunities to reinvent itself and contribute even more to societal well-being [60]. As noted earlier, Frost, Mair, and Laing (2014) identify three educational opportunities in greening events: to raise awareness, to encourage behavior change as part of a larger campaign, and to use the festival or event to play an advocacy role [34] (see also [32]). Such pro-environmental action is integral to destination resilience in the learning destination.

Future considerations for sustainable event frameworks such as the SEC could therefore include adding criteria on community and visitor engagement, literacy, education, and awareness raising to the sustainable event checklists as important social dimensions. These are valuable characteristics of the learning destination, contributing to an informed public able to influence environmental policy and contribute to destination resilience.

A valuable paradigm thus arises for destinations striving for resilience and sustainability, including the sustainability of its events and their resilience to future threats and emergent challenges. The learning destination case presented in this paper offers a first-hand look at its sustainability initiatives and the micro-level, practical actions undertaken to develop the SEC through the participatory action and involvement of students, staff, faculty, and community stakeholders (e.g., event organizers).

One limitation of the case presented here is that greater attention needed to be paid to visitor engagement and awareness raising. Additionally, while the case generally corroborates Sadd et al.'s (2017) characterization of the learning destination, future research (quantitative and qualitative) is needed to closely examine the process, goals, outcomes,

the general and specific criteria for resilience in the learning destination, and sustainability in the event domain [10].

Lastly, multi-stakeholder and community-based experiential learning approaches (including the monitoring and evaluation of experiential education and service-learning approaches) are highly recommended in the learning destination. This would focus not just on student learning but also on community and institutional changes in efforts to advance resilience and sustainability in the learning destination and event domain. As Paschal et al.'s (2019, p. 43) study of service learning in a four-year public university indicates, "opportunities arise for discovery of self and others, and the impacts of the liminal space of service-learning transcending traditional academic boundaries."

## 7. Conclusions

Significant shifts occurred in the event management domain when COVID-19 paralyzed global to local economies in 2020. As destinations began to emerge cautiously in late 2021 and 2022, it was clear that new approaches, considerations, and literacies are needed to address future crises, as well as some systemic changes (e.g., greatly increased online presence by service providers and participation by consumers). The Special Issue called contributors to "reimagine how events will be designed, delivered, supported, and evaluated in the future, and their value to individuals, organizations, and destinations" (https://www.mdpi.com/journal/tourismhosp/special_issues/festival_events, accessed on 9 January 2023). This collaborative paper has drawn on the interdisciplinary expertise and experience of its authors to propose one helpful approach toward destination resilience and disaster management: a place-based, situated approach to addressing destination resilience and event sustainability through the paradigmatic lens of a learning destination (Sadd et al.'s (2017) concept [10]).

As we have learned from the pandemic, preparing a destination and its event domain to be resilient and responsive to emergent issues, future shocks, and disasters requires proactive planning, learning, and adaptive skills to handle complex domains. We argued early on that the learning destination in which recreational and leisure events are enacted to serve local residents and the visiting public is a complex social–ecological system containing multiple stakeholders with different agendas; the events being enacted can have economic, ecological, and social–cultural as well as political impacts, and the place is subject to external and internal threats, including climate change. Therefore, event managers must be capable of engaging in collaborative learning and adaptive planning, building knowledge and skills to address emergent issues and wicked problems, implementing sustainable event practices and management actions, and hence contributing to destination resilience.

This paper identifies some general criteria, literacies, and lessons to inform destination resilience and event sustainability as we move forward into the post-pandemic Anthropocene. The learning destination paradigm presented here is a university that is an educational destination and a sport, recreational, and leisure destination, hosting a large number of sporting and social events throughout the year. The characteristics of the learning destination and the processes, actions, literacies, and skills developed to facilitate destination resilience and event sustainability within a circular economy are described in the case and summarized in Figure 1 and Table 2. This offers a valuable paradigm for destinations hosting events, which are complex systems by definition [38]. Collaboration, adaptation, continuous learning, and skill building (including soft skills) undertaken in the sustainable event domain contribute to overall place resilience and preparedness to handle emergent challenges and threats.

The case also shows how the TAMU learning institution collaboratively developed the SEC, a certification framework to help event stakeholders incorporate all areas of sustainability into their events. Sadd et al. (2017), in fact, argue that "to be truly beneficial and to be a true learning destination, evaluation frameworks . . . need to be implemented in a truly inclusive, integrated, and collaborative manner for objective strategic decision making to be effective and for destination wide interaction and engagement to exist and,

more importantly, to be sustained" [10] (p. 347). An ethic of collaborative engagement among stakeholders facilitates knowledge sharing and continuous learning, helping spur innovation and action in the event domain and sustain the broader destination as it builds resilience and essential qualities for preparedness to face future threats and issues as they arise.

In conclusion, collaborative learning, engagement, and continuous improvement are crucial for event-based destinations to respond swiftly to emergent challenges and wicked problems such as climate change. Future research needs to more closely address the political and policy processes that influence destination sustainability while also impacting the sustainability of events within the domain and adaptive planning and preparedness for future crises by the multiple stakeholders in the complex learning destination. As Sadd et al. (2017) suggest, future research should also undertake qualitative interviews with salient stakeholders and quantitative studies to advance knowledge building on the learning destination [10]. Attention should be paid to difficult-to-measure and, therefore, easy-to-miss aspects, such as building soft skills and practical knowledge (practical wisdom, *phronêsis*) to make difficult choices and identify and enact the trade-offs sometimes needed to advance event sustainability and destination resilience.

**Author Contributions:** All authors contributed equally to the study. The authors are listed in alphabetical order. J.C. and K.W. wrote up the Texas A&M university-level sustainability activities and events framework (SEC), which they were directly involved in developing. They also assisted with editing. S.L. and T.J. undertook the literature review and contributed to the conceptual framing and analysis, writing, editing, and formatting. D.L.S. provided the case example of event management students in her classes, who worked with the SEC Checklist and assisted with editing. All authors have read and agreed to the published version of the manuscript.

**Funding:** Partial funding for the pilot testing by the undergraduate event classes was received from an internal grant in 2020, the Innovation [X] grant for the project "Innovation for Sustainability: Outlining Opportunities to Transition Texas A&M to Carbon Neutrality".

**Informed Consent Statement:** All students enrolled in the courses described, the subjects, provided written informed consent. The Texas A&M University Office of Academic Innovation authorized the creation of FERPA release forms in response to inquiry of the ethics in releasing student educational records that depicted and described their event management classroom work.

**Acknowledgments:** We thank the Texas A&M University, School of Innovation (now transitioned to LAUNCH, a division of Undergraduate Studies) for the 2020 Innovation [X] grant, which contributed some funds to involve the event management classes in pilot testing the Sustainable Event Certification in 2021–2022.

**Conflicts of Interest:** The authors declare that there is no conflict of interest.

**Appendix A. Local Vendors: Food**

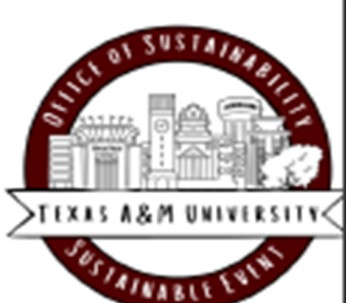

LOCAL VENDORS
FOOD

| | |
|---|---|
| 40 Tempura | Burger Mojo |
| 555 Grill | C&C's Asian Garden |
| 1541 Pastries and Coffee | C&J BBQ |
| 1775 Texas Pit BBQ | Caffe Capri |
| 1860 Italia | Carport Coffee |
| 3rd on Main | Casa Rodriguez |
| Aji Sushi | Centro American Restaurant & Pupuseria |
| All the King's Men | Chef Cao's |
| Amico Nave Ristorante | Choi's Restaurant |
| Babe's Doughnut & Coffee Shop | Coco Loco |
| Babylon Cafe | Crawfish Hole |
| Baked or Fried | DBQ Barbecue & Catering |
| Bangkok Thai Kitchen | Dixie Chicken |
| Bavarian Brauhaus | Don Chente |
| BonAppeTea | Fusion Peru |
| Burger House | |

# LOCAL VENDORS

## FOOD

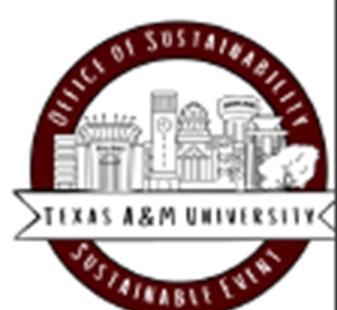

| | |
|---|---|
| Garpez | Nam Cafe |
| Gumby's Pizza | Napa Flats |
| I PHO | Ohana Korean Grill |
| Jesse's Taqueria | Oishi Sushi Asian Fusion |
| Kai's Donuts | Paolo's Italian Kitchen |
| Kamei Sushi & Grill | Papa Perez |
| Kolache Rolf's | Pepe's Mexican Cafe |
| JJ's Snowcones | Piasano's Italian Pizzeria |
| La Botana | Poke Stop |
| La Familia Taqueria | Polite Coffee Roasters |
| La Gabriella Coffee and Pastries | Polly's Cocina |
| Lamar & Niki's | Porters Dining |
| Luigi's Patio Ristorante | Proudest Monkey |
| Mess Waffles | Readfield's Meats & Deli |
| Mong Chon | Rooster's Bike and Coffee Shop |
| Mr G's Pizzeria | Ronin |

# LOCAL VENDORS

## FOOD

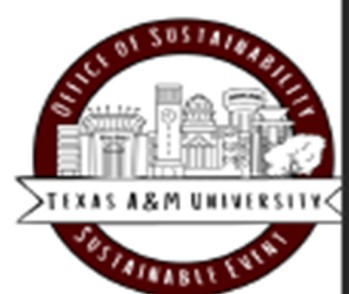

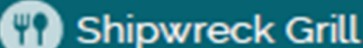 Shipwreck Grill

The Corner Bar and Grill 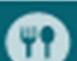

Shiraz Shish Kabob

The Republic Steakhouse 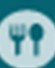

Shun de Mom Express

The Spot on Northgate 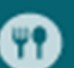

Stella Southern Cafe

The Tacobar 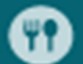

Sweet Eugene's

The Village 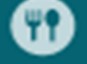

Sweet Relish Cafe & Baked Goods

The Wild Garlic 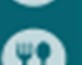

Tacos La Perlita

Three Daughters Chocolates 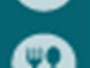

Taco Crave

Truman Chocolates 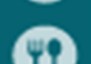

Taqueria Poblana

Twisted Noodle Cafe 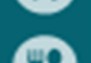

Taz Indian Cuisine

Urban Table 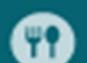

The Brew Coffeehouse

Yole's La Familia Taqueria 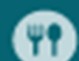

The Chocolate Gallery

Zand's Persian Kebabs

Zeitman's Grocery Store & Deli 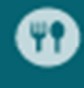

**Appendix B. Local Vendors: Floral**

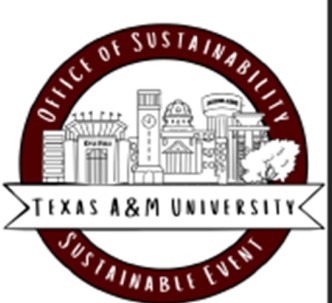

# LOCAL VENDORS
# FLORAL

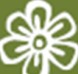

Aggieland Flowers and Chocolates

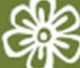

My Little Bee Flowers and More

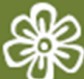

Tricia Barksdale Designs

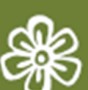

Angelle's Floral LLC

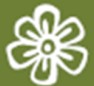

Nan's Blossom Shop

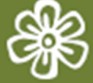

University Flowers

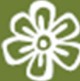

Busy Lil Bee Floral

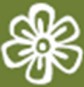

Nita's Flowers Inc.

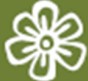

Urban Rubbish Floristry

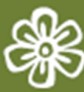

Down to Earth Florals

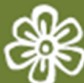

Petal Patch Florist

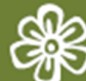

Wild Blue Iris

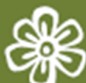

Flawless Florals

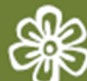

Postoak Florist

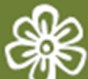

Willow Lane Florals

**Appendix C. Local Vendors: Rentals**

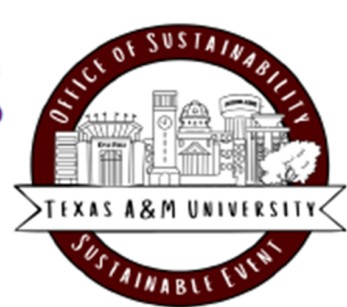

LOCAL VENDORS
RENTALS

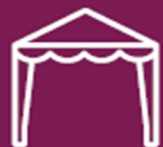

Ashley and
Company

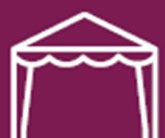

Alpha-Lit
Bryan College
Station

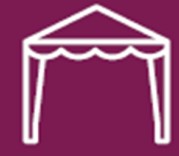

Card My
Yard

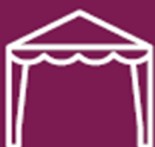

Details Party
Rental

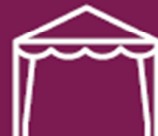

Events to
Remember

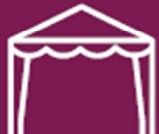

Premiere
Events

**Appendix D. Sustainable Buildings**

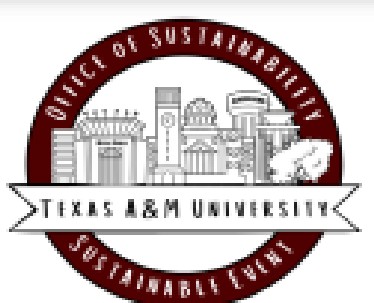

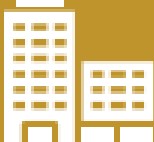

Department of Multicultural Services

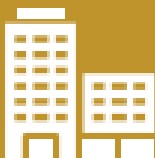

Memorial Student Center

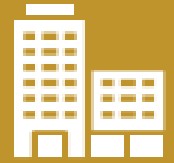

Innovative Learning Classroom Building

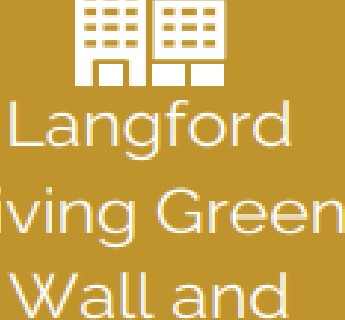

Langford Living Green Wall and Green Roof

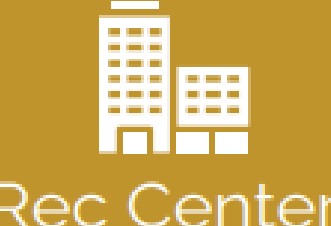

Rec Center

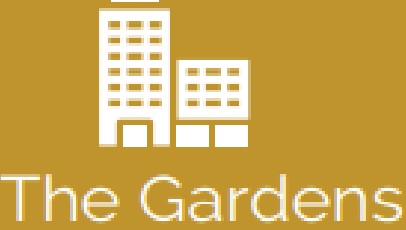

The Gardens

**Appendix E. Event Signage**

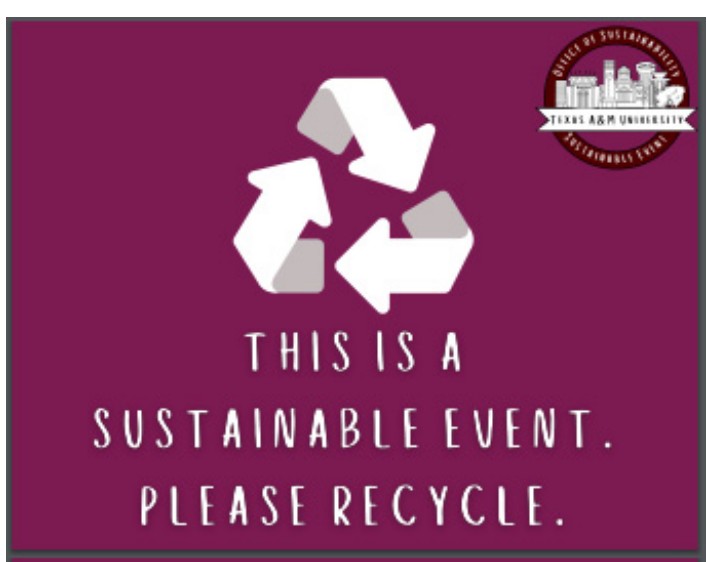

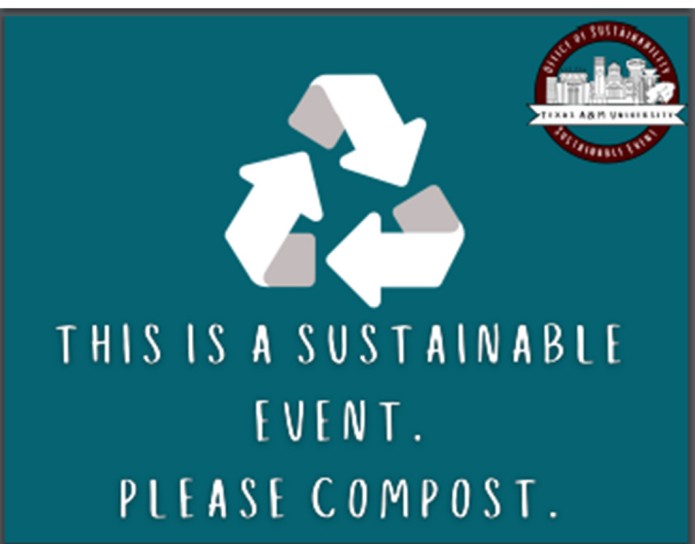

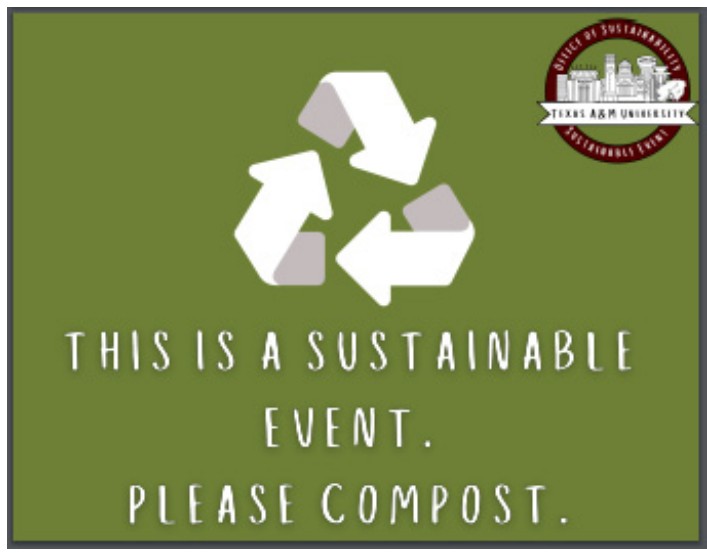

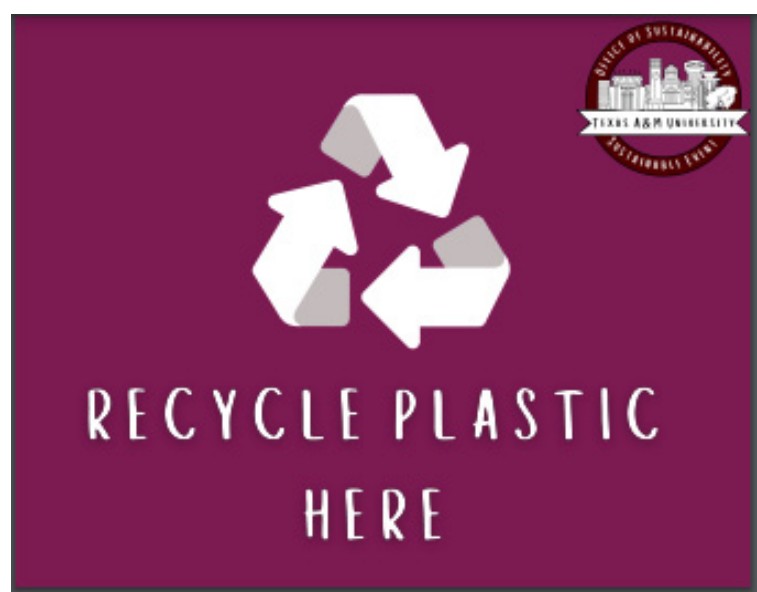

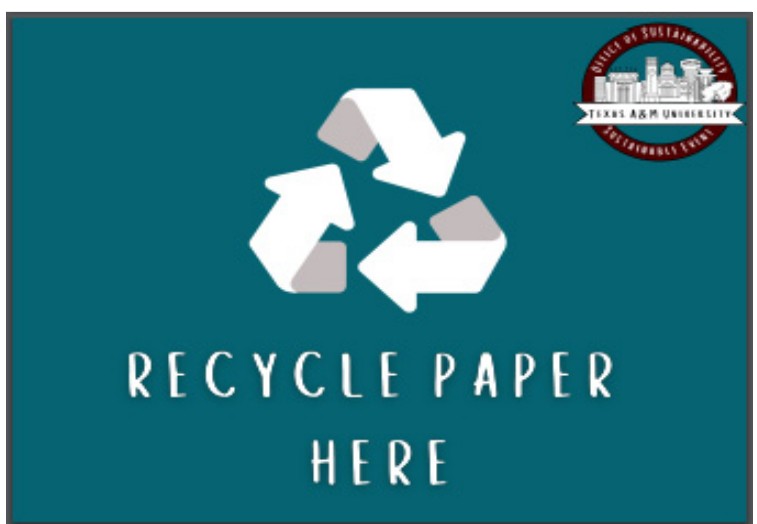

**Appendix F. TAMU Sustainable Event Certification Checklist**

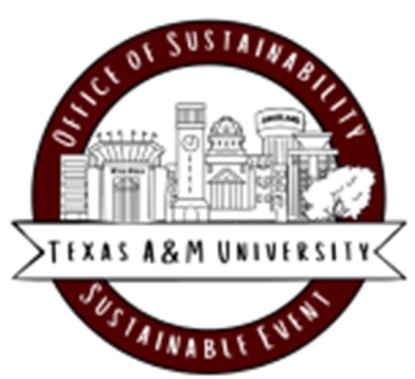

**Step 1:** Please review the entire checklist prior to beginning event planning.
**Step 2:** Complete the checklist to the best of your availability no later than two weeks from your event.
**Step 3:** Once you have completed the entire form, please tally your score and put it in the total points earned section at the bottom of the checklist.
**Step 4:** Return the checklist and any additional information to sustainability@tamu.edu, subject: Sustainable Event Certification. A representative from the office will review your submission and contact you.

### PRIMARY CONTACT INFORMATION

Name: ________________________________________ Choose One: ________

Department/Organization: ___________________ Phone Number: _______________

Email Address: ________________________________________________________

### SECONDARY CONTACT INFORMATION

Name: ________________________________________ Choose One: ________

Department/Organization: ___________________ Phone Number: _______________

Email Address: ________________________________________________________

### EVENT INFORMATION

Event Name: ___________________________________________________________

Date: ______________ Time: ________________ # of Attendees: _____________

Event Location: _____________________ URL (if available): _________________

Brief Description of Event:

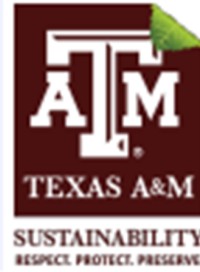

**Please check all of the items that apply to your event.**

## PURCHASING

- ☑ Earth friendly, bio-degradable cleaning products will be used.
- ☑ Sustainable centerpieces will be used and/or natural décor will be used.
- ☑ Sustainable centerpieces will be raffled off or given away so not to end up in landfill.
- ☑ Floral or plant arrangements will be locally sourced and in season (see Local Vendors: Floral).
- ☑ Décor will be reusable when possible and saved for future events.
- ☑ Any event giveaways will promote sustainable practices.
- ☑ Necessary items (chairs, tables, etc.) will be borrowed from campus partners and university departments or rented before purchasing new supplies (See Local Vendors: Rentals).
- ☑ Items that need to be purchased will be done so locally when possible.
- ☑ Any thank you notes will be electronic. Any thank you gifts will be sustainable or consumable.
- ☑ T-shirts will be made from recycled materials or organic cotton and/or purchased from Historically Underutilized Business (HUB) Vendors. HUB Vendors can be found here.

*Please describe any checkmarks below:*

**Points Earned: _______________ Points Available: 10**

## WASTE REDUCTION

- ☑ Landfill and recycling bins will be provided to minimize the amount of waste going to a landfill.
- ☑ Signs for recycling will be placed around the event and bins will be labeled (See Event Signage).
- ☑ Food will be served that does not require dining ware or utensils. If food is served that requires dining ware, reusable dining ware or utensils will be utilized.
- ☑ The event will use reusable napkins or no napkins at all.
- ☑ The event will use pitchers or water coolers for drinks, single use plastic bottles will not be purchased.
- ☑ Attendees will be encouraged to bring their own reusable mug/cup/bottle.
- ☑ The event will use reusable tablecloths or no tablecloths at all.
- ☑ The event will provide a compost bin and dispose of waste properly.
- ☑ The event will eliminate individually wrapped condiments, sugar, salt, pepper, creamer, etc.
- ☑ The event will eliminate plastic coffee stirrers, straws, single use lids, etc.
- ☑ 100% biodegradable garbage bags will be used to collect waste and will be disposed of properly.
- ☑ This even will use eusable name badges that will be collected at the end of the event or no name badges will be used.

*Please describe any checkmarks below:*

**Points Earned: _______________ Points Available: 12**

**Please check all of the items that apply to your event.**

## TRANSPORTATION & LOCATION

- ☑ The event will be held completely virtually or have a virtual option.
- ☑ Event will be held on campus to minimize transportation emissions.
- ☑ Event will be held in a sustainable building (See Sustainable Buildings) or outside.
- ☑ Alternative travel options will be available and information will be shared prior to the event.
- ☑ Carpooling and use of alternative transportation will be encouraged.
- ☑ If the event will be held off campus, it is held at a location convenient for public transportation, biking, or walking. Learn more about Brazos Transit District here.
- ☑ If hotel accommodations are required, guests will be encouraged to stay in a central area so that alternative modes of transportation and carpooling can be used.
- ☑ If hotel accommodations are required, guests will be encouraged to reuse towels, unplug personal electronics when not in use, adjust room temperatures while gone, bring their own toiletries in refillable bottles, etc.

*Please describe any checkmarks below:*

**Points Earned:** _______________ **Points Available: 8**

## SOCIAL SUSTAINABILITY

- ☑ The event will be focused on environmental, multicultural, and/or equity related topics.
- ☑ Attendees will be encouraged to donate to a non-profit or community organization at the event (e.g. canned food drive, clothing drive, monetary donation, etc.)
- ☑ A land acknowledgment will open the event. A sample land acknowledgement can be found here.
- ☑ A physical activity to encourage attendees to move for better health will be included.
- ☑ The event will be wheelchair/ADA accessible.
- ☑ Communications for the event will be available in different languages.
- ☑ The event will raise awareness through speakers or panels on prominent social issues.
- ☑ Organizations and departments such as Aggie Allies, Green Dot, LGBTQ+ Pride Center, Women's Resource Center, etc. will be partners of the event.

*Please describe any checkmarks below:*

**Points Earned:** _______________ **Points Available: 8**

**Total Points Earned:** _______________ **out of 61 =** _______________ **%**

**Appendix G. Annual Sustainability Awards Ceremony 2022 Checklist**

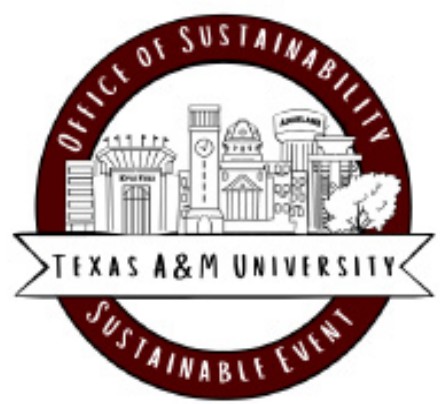

**Step 1:** Please review the entire checklist prior to beginning event planning.

**Step 2:** Complete the checklist to the best of your availability no later than two weeks from your event.

**Step 3:** Once you have completed the entire form, please tally your score and put it in the total points earned section at the bottom of the checklist.

**Step 4:** Return the checklist and any additional information to sustainability@tamu.edu, subject: Sustainable Event Certification. A representative from the office will review your submission and contact you.

## PRIMARY CONTACT INFORMATION

Name: Jesse Carswell                                                   Choose One: Staff

Department/Organization: Office of Sustainability          Phone Number: 2543193249

Email Address: jcarswell@tamu.edu

## SECONDARY CONTACT INFORMATION

Name: Kelly Wellman                                                   Choose One: Staff

Department/Organization: Office of Sustainability          Phone Number: 979-845-1911

Email Address: kwellman@tamu.edu

## EVENT INFORMATION

Event Name: Sustainability Awards Breakfast

Date: 4/22/2022                 Time: 8:30 a.m.                 # of Attendees: <100

Event Location: MSC 2400 Gates Ballroom          URL (if available):

Brief Description of Event:

Annual Office of Sustainability Awards Ceremony and Breakfast. STARS and Sustainability Champions will be announced.

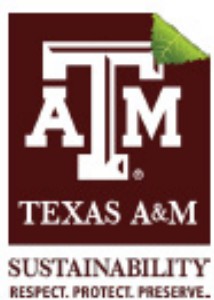

**Please check all of the items that apply to your event.**

## PLANNING

- ☑ To get a more accurate headcount and minimize food waste, attendees must RSVP to the event.
- ☑ Dietary needs will be asked prior to the event and event team will ensure food labels are provided.
- ☑ There will be a plan for food leftovers (compost, donation, packed and taken to eat later).
- ☑ Dates and slogans will not be printed on banners, signs, and posters for recurring events.
- ☑ The event will be recorded and made publicly available for those unable to attend.
- ☑ The event will include a presentation on campus, local topics, or initiatives, etc.
- ☐ If the presenter is not local (100+ miles), they will present virtually to minimize emissions and costs.
- ☑ The event team set zero waste goals for the event.

*Please describe any checkmarks below:*

RSVPs were collected and in the form, dietary restrictions were asked about. Reusable banners with no dates will be used at the event. It will be recorded and livestreamed.

Points Earned: 7 ________ Points Available: 8

## PROMOTION

- ☑ All promotion for the event will be done digitally / paperless.
- ☑ When advertising digitally, media will be accessible to people with screen readers.
- ☑ Social media or campus emails will be the primary source of promotion.
- ☑ Using paper handouts for promotion will be minimized.
- ☑ If paper promotion occurs, scrap, reused, or 30%+ recyclable paper will be utilized.
- ☑ The event team will print multiple flyers on one sheet of paper rather than full page flyers.
- ☐ Advertisements utilize reusable sandwich boards and/or yard signs.
- ☐ The team advertises using reusable bus ads.

*Please describe any checkmarks below:*

Handouts will only be utilized once for the event to let attendees know codewords. Will be on half sheets printed on recycled paper. All promotion otherwise will be digital/social media.

Points Earned: 6 ________ Points Available: 8

## FOOD

- ☑ Vegetarian and/or vegan food options will be available if the event will have food.
- ☑ Local vendors will be used for catering (See Local Vendors: Food).
- ☑ To minimize packaging, food will be served buffet style.
- ☑ Fair Trade certified snacks and beverages will be served. Learn more about Fair Trade here.
- ☑ To reduce carbon footprint, food options will minimize meat options.
- ☐ Food cultures from different regions will be celebrated and served.
- ☑ Produce served will be local and in season.

*Please describe any checkmarks below:*

Vegetarian options will be served by Aggie Dining served buffet style. Fair Trade coffee will be served and produce will be in season.

Points Earned: 6 ________ Points Available: 7

**Please check all of the items that apply to your event.**

## PURCHASING

- ☑ Earth friendly, bio-degradable cleaning products will be used.
- ☑ Sustainable centerpieces will be used and/or natural décor will be used.
- ☑ Sustainable centerpieces will be raffled off or given away so not to end up in landfill.
- ☑ Floral or plant arrangements will be locally sourced and in season (see Local Vendors: Floral).
- ☑ Décor will be reusable when possible and saved for future events.
- ☑ Any event giveaways will promote sustainable practices.
- ☑ Necessary items (chairs, tables, etc.) will be borrowed from campus partners and university departments or rented before purchasing new supplies.
- ☑ Items that need to be purchased will be done so locally when possible.
- ☐ Any thank you notes will be electronic. Any thank you gifts will be sustainable or consumable.
- ☐ T-shirts will be made from recycled materials or organic cotton and/or purchased from Historically Underutilized Business (HUB) Vendors. HUB Vendors can be found here.

*Please describe any checkmarks below:*

Centerpieces will be produce from TUHF and will be given away. All necessary items will be provided by the MSC.

Points Earned: **8** Points Available: 10

## WASTE REDUCTION

- ☑ Landfill and recycling bins will be provided to minimize the amount of waste going to a landfill.
- ☑ Signs for recycling will be placed around the event and bins will be labeled (See Recycling Signs).
- ☑ Food will be served that does not require dining ware or utensils. If food is served that requires dining ware, reusable dining ware or utensils will be utilized.
- ☑ The event will use reusable napkins or no napkins at all.
- ☑ The event will use pitchers or water coolers for drinks, single use plastic bottles will not be purchased.
- ☐ Attendees will be encouraged to bring their own reusable mug/cup/bottle.
- ☑ The event will use reusable tablecloths or no tablecloths at all.
- ☐ The event will provide a compost bin and dispose of waste properly.
- ☑ The event will eliminate individually wrapped condiments, sugar, salt, pepper, creamer, etc.
- ☑ The event will eliminate plastic coffee stirrers, straws, single use lids, etc.
- ☐ 100% biodegradable garbage bags will be used to collect waste and will be disposed of properly.
- ☑ This even will use eusable name badges that will be collected at the end of the event or no name badges will be used.

*Please describe any checkmarks below:*

Recycling signage and bins will be available and reusable dining items will be used. No name badges will be used.

Points Earned: **9** Points Available: 12

**Please check all of the items that apply to your event.**

## TRANSPORTATION& LOCATION

- ☑ The event will be held completely virtually or have a virtual option.
- ☑ Event will be held on campus to minimize transportation emissions.
- ☑ Event will be held in a sustainable building (See Sustainable Buildings).
- ☐ Alternative travel options will be available and information will be shared prior to the event.
- ☑ Carpooling and use of alternative transportation will be encouraged.
- ☐ If the event will be held off campus, it is held at a location convenient for public transportation, biking, or walking. Learn more about Brazos Transit District here.
- ☐ If hotel accommodations are required, guests will be encouraged to stay in a central area so that alternative modes of transportation and carpooling can be used.
- ☐ If hotel accommodations are required, guests will be encouraged to reuse towels, unplug personal electronics when not in use, adjust room temperatures while gone, bring their own toiletries in refillable bottles, etc.

*Please describe any checkmarks below:*

**The event will be livestreamed and be held in the MSC.**

**Points Earned: 4**     **Points Available: 8**

## SOCIAL SUSTAINABILITY

- ☑ The event will be focused on environmental, multicultural, and/or equity related topics.
- ☐ Attendees will be encouraged to donate to a non-profit or community organization at the event (e.g. canned food drive, clothing drive, monetary donation, etc.) (See Community Organizations)
- ☑ A land acknowledgment will open the event. A sample land acknowledgement can be found here.
- ☐ A physical activity to encourage attendees to move for better health will be included.
- ☑ The event will be wheelchair/ADA accessible.
- ☐ Communications for the event will be available in different languages.
- ☑ The event will raise awareness through speakers or panels on prominent social issues.
- ☐ Organizations and departments such as Aggie Allies, Green Dot, LGBTQ+ Pride Center, Women's Resource Center, etc. will be partners of the event.

*Please describe any checkmarks below:*

**The event will be focused on campus sustainability.**

**Points Earned: 4**     **Points Available: 8**

**Total Points Earned: 44 + 6 = 50**    **out of 61 = 81.9**    **%**

### Appendix H. Campus Sustainability Day 2022 Checklist

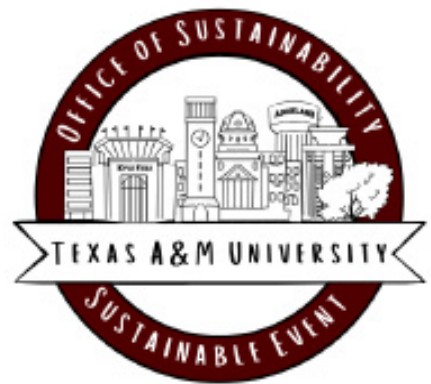

**Step 1:** Please review the entire checklist prior to beginning event planning.

**Step 2:** Complete the checklist to the best of your availability no later than two weeks from your event.

**Step 3:** Once you have completed the entire form, please tally your score and put it in the total points earned section at the bottom of the checklist.

**Step 4:** Return the checklist and any additional information to sustainability@tamu.edu, subject: Sustainable Event Certification. A representative from the office will review your submission and contact you.

#### PRIMARY CONTACT INFORMATION

Name: Jesse Carswell                          Choose One: Staff

Department/Organization: Office of Sustainability          Phone Number: 9794588112

Email Address: jcarswell@tamu.edu

#### SECONDARY CONTACT INFORMATION

Name: _______________________          Choose One: _______________

Department/Organization: _______________          Phone Number: _______________

Email Address: _______________

#### EVENT INFORMATION

Event Name: Campus Sustainability Day

Date: 10/19/2022          Time: 10 a.m. - 1 p.m.          # of Attendees: 250+

Event Location: Rudder Plaza          URL (if available): https://sustainability.tamu

**Brief Description of Event:**

Campus Sustainability Day will take place on Wednesday, October 19 from 10 a.m. to 1 p.m. in Rudder Plaza. It will feature over 20 organizations that represent student groups, community partners, and a variety of TAMU departments that have implemented sustainable initiatives.

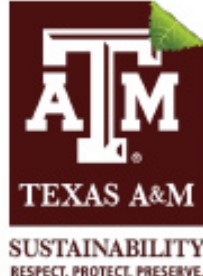

**Please check all of the items that apply to your event.**

## PLANNING

- ☐ To get a more accurate headcount and minimize food waste, attendees must RSVP to the event.
- ☐ Dietary needs will be asked prior to the event and event team will ensure food labels are provided.
- ☐ There will be a plan for food leftovers (compost, donation, packed and taken to eat later).
- ☑ Dates and slogans will not be printed on banners, signs, and posters for recurring events.
- ☐ The event will be recorded and made publicly available for those unable to attend.
- ☑ The event will include a presentation on campus, local topics, or initiatives, etc.
- ☑ If the presenter is not local (100+ miles), they will present virtually to minimize emissions and costs.
- ☐ The event team set zero waste goals for the event.

*Please describe any checkmarks below:*

**All presenters will local organizations, departments, or partners.**

**Points Earned: 3**      **Points Available: 8**

## PROMOTION

- ☑ All promotion for the event will be done digitally / paperless.
- ☑ When advertising digitally, media will be accessible to people with screen readers.
- ☑ Social media or campus emails will be the primary source of promotion.
- ☑ Using paper handouts for promotion will be minimized.
- ☑ If paper promotion occurs, scrap, reused, or 30%+ recyclable paper will be utilized.
- ☑ The event team will print multiple flyers on one sheet of paper rather than full page flyers.
- ☑ Advertisements utilize reusable sandwich boards and/or yard signs.
- ☐ The team advertises using reusable bus ads.

*Please describe any checkmarks below:*

**Promotion will be done digitally and through social media. The only paper that was used was passports which will be printed on recycled paper and four to a page. Reusable sandwich boards and flags will be used day of.** ➕

**Points Earned: 7**      **Points Available: 8**

## FOOD

- ☑ Vegetarian and/or vegan food options will be available if the event will have food.
- ☐ Local vendors will be used for catering (See Local Vendors: Food).
- ☐ To minimize packaging, food will be served buffet style.
- ☑ Fair Trade certified snacks and beverages will be served. Learn more about Fair Trade here.
- ☑ To reduce carbon footprint, food options will minimize meat options.
- ☐ Food cultures from different regions will be celebrated and served.
- ☐ Produce served will be local and in season.

*Please describe any checkmarks below:*

**Vegetarian snacks will be provided by several groups. Fair Trade products will be provided by Aggie Dining.**

**Points Earned: 3**      **Points Available: 7**

**Please check all of the items that apply to your event.**

## PURCHASING

- ☑ Earth friendly, bio-degradable cleaning products will be used.
- ☐ Sustainable centerpieces will be used and/or natural décor will be used.
- ☐ Sustainable centerpieces will be raffled off or given away so not to end up in landfill.
- ☐ Floral or plant arrangements will be locally sourced and in season (see Local Vendors: Floral).
- ☑ Décor will be reusable when possible and saved for future events.
- ☑ Any event giveaways will promote sustainable practices.
- ☑ Necessary items (chairs, tables, etc.) will be borrowed from campus partners and university departments or rented before purchasing new supplies (See Local Vendors: Rentals).
- ☑ Items that need to be purchased will be done so locally when possible.
- ☑ Any thank you notes will be electronic. Any thank you gifts will be sustainable or consumable.
- ☑ T-shirts will be made from recycled materials or organic cotton and/or purchased from Historically Underutilized Business (HUB) Vendors. HUB Vendors can be found here.

*Please describe any checkmarks below:*

> **Event giveaways are those for the entire month giveaway and all promote sustainability. Tables and chairs will be rented from the MSC. Thank yous will be sent after the event. T-shirts were made from local, woman owned business.**

**Points Earned:** _7_ **Points Available: 10**

## WASTE REDUCTION

- ☐ Landfill and recycling bins will be provided to minimize the amount of waste going to a landfill.
- ☐ Signs for recycling will be placed around the event and bins will be labeled (See Event Signage).
- ☐ Food will be served that does not require dining ware or utensils. If food is served that requires dining ware, reusable dining ware or utensils will be utilized.
- ☑ The event will use reusable napkins or no napkins at all.
- ☑ The event will use pitchers or water coolers for drinks, single use plastic bottles will not be purchased.
- ☐ Attendees will be encouraged to bring their own reusable mug/cup/bottle.
- ☑ The event will use reusable tablecloths or no tablecloths at all.
- ☐ The event will provide a compost bin and dispose of waste properly.
- ☑ The event will eliminate individually wrapped condiments, sugar, salt, pepper, creamer, etc.
- ☑ The event will eliminate plastic coffee stirrers, straws, single use lids, etc.
- ☐ 100% biodegradable garbage bags will be used to collect waste and will be disposed of properly.
- ☑ This even will use eusable name badges that will be collected at the end of the event or no name badges will be used.

*Please describe any checkmarks below:*

> **The event will not be using single use products.**

**Points Earned:** _6_ **Points Available: 12**

**Please check all of the items that apply to your event.**

## TRANSPORTATION & LOCATION

- ☐ The event will be held completely virtually or have a virtual option.
- ☑ Event will be held on campus to minimize transportation emissions.
- ☑ Event will be held in a sustainable building (See Sustainable Buildings) or outside.
- ☑ Alternative travel options will be available and information will be shared prior to the event.
- ☑ Carpooling and use of alternative transportation will be encouraged.
- ☐ If the event will be held off campus, it is held at a location convenient for public transportation, biking, or walking. Learn more about Brazos Transit District here.
- ☐ If hotel accommodations are required, guests will be encouraged to stay in a central area so that alternative modes of transportation and carpooling can be used.
- ☐ If hotel accommodations are required, guests will be encouraged to reuse towels, unplug personal electronics when not in use, adjust room temperatures while gone, bring their own toiletries in refillable bottles, etc.

*Please describe any checkmarks below:*

**Event will be held at Rudder plaza.**

**Points Earned: 4**_______ **Points Available: 8**

## SOCIAL SUSTAINABILITY

- ☑ The event will be focused on environmental, multicultural, and/or equity related topics.
- ☐ Attendees will be encouraged to donate to a non-profit or community organization at the event (e.g. canned food drive, clothing drive, monetary donation, etc.)
- ☐ A land acknowledgment will open the event. A sample land acknowledgement can be found here.
- ☐ A physical activity to encourage attendees to move for better health will be included.
- ☑ The event will be wheelchair/ADA accessible.
- ☐ Communications for the event will be available in different languages.
- ☑ The event will raise awareness through speakers or panels on prominent social issues.
- ☑ Organizations and departments such as Aggie Allies, Green Dot, LGBTQ+ Pride Center, Women's Resource Center, etc. will be partners of the event.

*Please describe any checkmarks below:*

**Points Earned: 4**_______ **Points Available: 8**
**Total Points Earned: 34**_______ **out of 61 = 55.7**_______ **%**

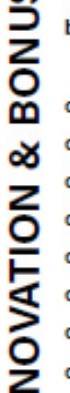

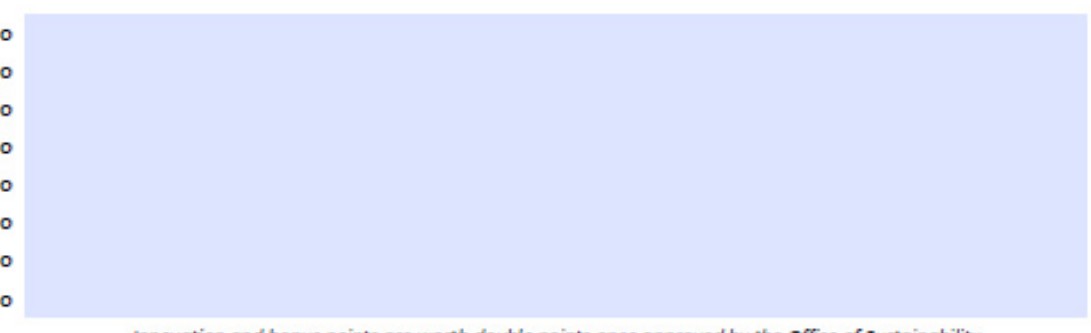

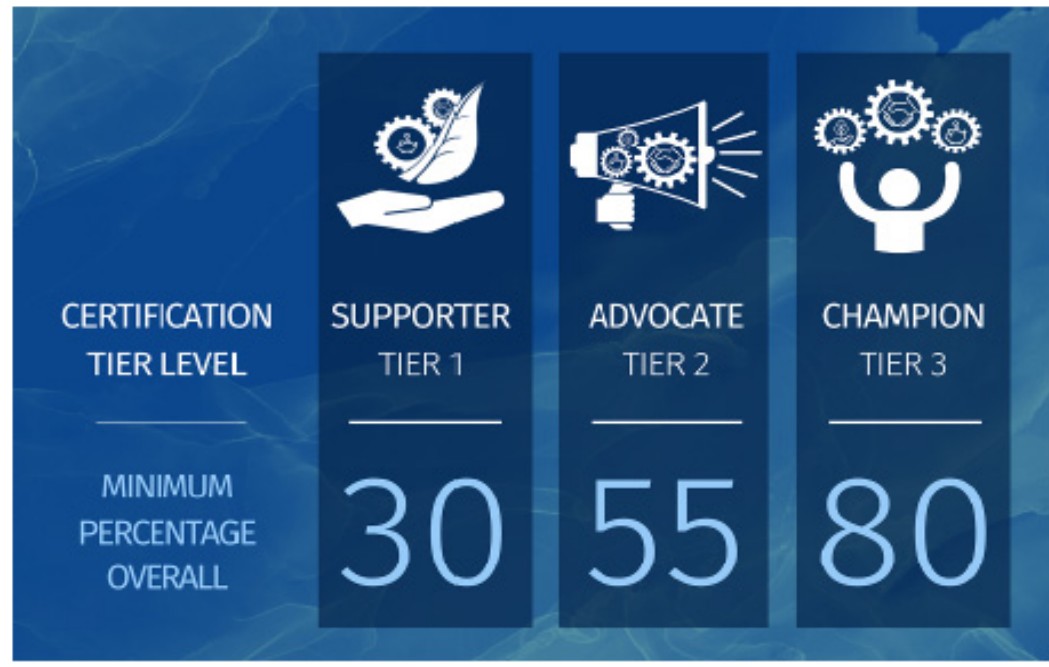

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
