# Peer review of "Post-Pandemic Lessons for Destination Resilience and Sustainable Event Management: The Complex Learning Destination"

_tourismhosp, doi:10.3390/tourhosp4010007_

Round 1

Reviewer 1 Report

Congratulations to the authors as the paper is well-written and makes a contribution to tourism events studies on the particular note that events are analysed in a learning destination and complex perspectives. A relevant literature review is developed and that is of great importance with examples related to green events, circular economies and pro-social events´ characteristics and experiences of participants and local communities.  

Also worth mentioning is that the paper is in line with the special issue it intends to allude to and contribute.

Key authors of event studies are mentioned and recent literature also.

Examples always accompany the discussion, pragmatic approaches are justified and the way the criteria of the complex approach to learning destinations are well presented. The implementation of the checklist is also clear.

Congratulations

Author Response

We are very pleased to see that Reviewer 1 has identified the key merits and contributions of our manuscript. The reviewer’s time and comments are greatly appreciated, thank you!

Reviewer 2 Report

 The manuscript is very extensive. Supplementary elements take up a lot of space. I don't quite understand what the intention was to include them in the manuscript. Of course, the volume of the manuscript can be a moot point. If ‘Tourism and hospitality’ journal is able to accept such a large text, it cannot be its fault.

However, the manuscript gives the impression of a low-scholar text.

Detailed remarks:

- first of all, the methodological section is missing - this is a key part of the scientific article;

- the scientific purpose of the manuscript is not clear;

- research hypotheses/questions are missing - this is crucial in scientific texts;

- some figures (mainly photos) do not add anything to the scientific discussion.

The manuscript also has its good points. Cited references are relevant. The topic is quite interesting. However, the manuscript needs major revisions to increase its scientific value.

Author Response

Detailed remarks:

- first of all, the methodological section is missing - this is a key part of the scientific article;

OUR RESPONSE: The paper was already very long and we identified using a service learning and experiential learning approach to the practical case study. We have elaborated on this approach in our revisions. The case study approach has helped to corroborate important characteristics and criteria of the learning destination and further strengthened it as noted below.

- the scientific purpose of the manuscript is not clear;

OUR RESPONSE: Our thanks to Reviewer 2 for identifying possible lack of clarity here. We have made some edits to ensure the paper’s scientific and practical purpose and aims are clear.

- research hypotheses/questions are missing - this is crucial in scientific texts;

OUR RESPONSE: Thank you for your observation, we completely understand (we were striving to make the manuscript as accessible as possible to guide practice as well as offer conceptual contributions). This is not a quantitative study so there are no hypotheses. It uses an experiential learning and service learning approach to accomplish the above aims. We have elaborated on this in the methodology section and included research questions in the manuscript.

- some figures (mainly photos) do not add anything to the scientific discussion.

OUR RESPONSE: The figures/photos offer illustrative guides for the journal’s audience and have been accepted by the other two reviewers. We would greatly appreciate being able to leave them in.

- The manuscript also has its good points. Cited references are relevant. The topic is quite interesting. However, the manuscript needs major revisions to increase its scientific value.

OUR RESPONSE: We would like to thank Reviewer 2 for the time and effort put in to review our paper and offer valuable comments on our manuscript. The manuscript provides both scientific value and practical post-pandemic lessons for resilience and sustainable event management in the learning destination. We have made succinct edits that we hope will clarify this and offer additional value to the journal’s audience.

Reviewer 3 Report

Dear Authors,

your paper is very interesting and I would like to use it (with your permission)  in my Event Management classes once it has been published!

To improve your paper, please consider following suggestions:

Abstract: This paper aims to share post-pandemic lessons for destination resilience and sustaining the ability of events. It offers a new perspective that reimagines the space and place of events as learning  destinations [1] enmeshed in complex systems. Complexity arises due to the interactions and interrelationships between numerous stakeholders, activities and events in the social-ecological destination system where boundaries are porous, and issues and actions from afar can impact the local. The case presented here describes the micro-level activities and actions undertaken to engage with destination resilience and sustainable event management and certification at a learning destination in Texas, USA. These situated efforts are shown (i) at the campus-wide level for the university and (ii) 

with the collaborative, learning-oriented activities undertaken by students in the event management classes to pilot test the Sustainable Event Certification Checklist that was developed. They

corroborate the general characteristics and criteria of the complex learning destination summarized 

in the paper, along with identifying and discussing the skills, literacies, and lessons learned to 

advance destination resilience and the sustainability of events. Participants in the learning destination draw on practical knowledge and develop soft skills to engage in adaptive planning proactively and collaboratively with other stakeholders to address emergent challenges and practical ....

Line 34: Events Tourism is one of the fastest growing sectors worldwide, offering considerable economic and social benefits locally and globally through........

Line 53: Split the sentence:

Research on the pandemic’s impacts is still in the early stages. However, greater attention will be needed to manage events in the context of infectious disease outbreaks as online to offline life resumes. However, online and virtual activities will continue to be essential aspects of event planning and marketing [8].

Line 60: They also note the need for more attention synergizing wider social, community, and individual resilience perspectives.

Line 70: A collaborative approach to learning how to address the challenges is needed. Every stakeholder can only resolve the complex issues and wicked problems that..

Line 75: Complexity arises due to the interactions and interrelationships. between numerous stakeholders, activities and events in the destination system where boundaries are porous. Consequently, issues and actions from afar can impact the locals.

Line 82: ...place-based approach to post...

Line 86: ...provide a practical case example...

Line 96:...It is a public university tasked with the public good and well-being. It is an educational and recreational destination, hosting numerous sports and social and cultural events. Actions and initiatives towards destination...

Line 107: The following section...

Line 109: ...which sets ...

Line 115:... the loss of tourism...

Line 118: ...and the use of marketing...

Line 120: Governments stepped in to help the event sector worldwide, recognizing its vital importance in sustaining local and regional economies and its social and psychological health values as social isolation progressed in 2020 and 2021.

Line 122:  Outdoor events received government support in many destinations, recognizing their importance in offering economic and social resilience and their potential to become super-spreading sources of infection and outbreaks [3]. In addition, financial and policy support was vital to enable proper guidelines and measures to address crowding, social distancing, masking, etc., as well as developing new marketing....

Line 129: However, numerous other issues and impacts arise in enacting and managing events.

Line 136: ....environmental pollution....

Line 136: Transportation is critical in infrastructure considerations, and its environmental impacts are significant.

Line 147: ..Olympic Games shows..

Line 171: Key stakeholders ranging from local businesses, residents, visitors, non-profit organizations, and various public sector participants engage collaboratively to assist in greening events, sustaining the destination, and facilitating place resilience, including its inhabitants' well-being..

Line 176: ...play key roles in facilitating learning..

Line 188: ...addresses local as well as global issues...

Line 191:  ...short-duration festivals from...

Line 196: They are also events fulfilling important...

Line 199: In addition, as Mair and Laing (2012) note, greening events can be an excellent opportunity to educate and change attitudes and behavior within the organisation and towards the natural environment .

Line 208: ....as part of a more extensive campaign...

Line 209: ...in the UK pride themselves on reaching out to ...

Line 217: ...economy is a green and healthy economy where sustainable development and well-being are achievable..

Line 2222: Applying the circular economy...

Line 230: ...for the conservation of local..

Line 233: ...grows the local community; ...

Line 234: ...with neutral carbon initiatives, serving foods that are fair trade and organic,...

Line 238: ...and issues in the event management...

Line 239: Building sustainability and resilience in the event-based destination also requires learning, adapting, and managing the multiple stakeholders and dynamic context in which issues arise...

Line 259: Attractions like events are not isolated; they are part of the complex destination system. Their sustainability and resilience are interrelated with the emergent and dynamic properties of the complex system and the situated practices of multiple stakeholders within the local system external to it...

Line 262: Therefore, strategies for sustainable event management should be tailored to the place rather than aspire towards universal norms, as Dredge and Whitford (2010) note...

Line 276: Greening is about progress, not perfection. Ahmad et al. (2013) note that sustainability does not happen with just a single event [29]. ...

Line 275: It also requires making trade-offs and decisions that may only check some of the boxes in a sustainability framework....

Line 277: ...action presents opportunities ...

Line 289: The professionals interviewed also
pointed to the need for practical skills (applied knowledge) and stronger cooperation between universities and enterprises [42].

Line 296:  Some important steps they describe are vital stakeholders coming together to identify objectives...

Line 299: ...development of an event..

Line 303: Stakeholder involvement is crucial in planning...

Line 305: events and developing policies...

Line 306: As noted earlier, visitors are also important stakeholders and should be involved through awareness raising and engagement in event sustainability. Wong et al. (2015) examined green policies and...

Line 311:  ...and a healthy environment. ...

Line 319:...the computerization and ...

Line 321: Such virtual adaptations are integral to the post-COVID paradigm shift to facilitate event sustainability and destination resilience. These authors advocate creating new and specific international standards for managing event competition sites in the event of a new pandemic..

Line 329: Surprisingly, despite its immense local to global significance and importance in the context of recreation, leisure and tourism, sustainability-related accreditation schemes for the event industry still need to catch up.

Line 334: However, certifications for individual professionals in sustainable event planning have been gaining traction (see globalgreenevents.org and https://www.mpi.org/education/certificate-programs/sustainable-event-strategist). ...

Line 341: Furthermore, different green levels and logos can also be created, so that the audience can identify the ecological level that the event obtained [47]....

Line 359: The following section...

Line 371: AMU is a destination for student learning and professional development and a hub for numerous recreational activities, including sports, cultural, and community event destinations...

Line 376: ..and is home to men’s and women’s basketball and volleyball teams....

Line 388: The university is an essential focal point for education, recreation, and entertainment, with a significant environmental impact...

Line 390: and lower greenhouse gas emissions.

Line 390: The campus Sustainability Master Plan (SMP) provides a blueprint to make measurable, attainable actions for the university. It focuses on nine themes and defines 16 evergreen goals and 47 targets over a 20-year timeline.

Line 396: TAMU's primary Greenhouse Gas reduction effort is to reduce the overall campus energy consumption.

Line 401: ..size of over 29.6 million..

Line 403: an aggressive building-level energy reduction program known as the ..

Line 407: ..and was an example of how sports..

Line 415: A new baseline should be calibrated regularly to encourage good stewardship. Building systems require constant monitoring. Otherwise, the behaviors..

Line 419: Table 1. The reduction...

Line 424: Communicating with the public about the need for and results of important GHG reduction work is vital in raising awareness. ..

Line 427: ...sustainability of sports initiatives...

Line 465: ..socially sustainable events around campus.

Line 468: With the number of events happening all over campus, many environmental, economic, and social impacts could be minimized or mitigated with ...

Line 470: Reviewing other schools across the nation, many had some sustainable event certification, and it confirmed that our university needed one, too.

Line 473: of greening events as sustainable learning destinations.

Line 480: The first development phase was to..

Line 481: Most people think of the Waste Minimization checklist items when they consider sustainable choices.

Line 482: Reminders to have recycling bins available, use digital and social promotion instead of paper to advertise the event, and source reusable..

Line 486: ..were the following checklist ...

Line 487: ...organizations with limited ..

Line 510: ..actions in the initial ...

Line 512: the event's purpose...

Line 515: The first three items a planner can review are determining whether the event will be promoted digitally and accessible through social media, bulk emails, and campus TV signage.

Line 519: This checklist encourages using reusable promotion forms such as sandwich boards, yard signs, flags, or bus ads that do not have date-specific information...

Line 524:  ...for events have the potential for significant sustainability impact, positive or negative...

Line 525: Offering vegetarian and vegan entree choices is environmentally friendly and includes dietary restrictions.

Line 529: It lets attendees choose what goes on their plates, which can minimize food waste.

Line 538: Due to limited transport and fertiliser usage, serving local and in-season produce has a much smaller environmental impact than imported or out-of-season produce.

Line 542: ...organizations with limited funding sources,...

Line 544: Second, supporting local and sustainable companies is a great way to exercise both environmental
and social responsibility, limiting transportation emissions and providing income to local community members.....

Line 58: ...their cleaning products...

Line 551: Crafting tablescapes and centrepieces from natural items can be inexpensive to create a welcoming look at banquets.

Line 560: ...or other promotional items...

Line 563: After an event, it is customary ...

Line 570:  ..reduce the waste heading to a landfill,..

Line 571: ..important, so attendees...

Line 578: Transportation emissions account for a large amount of GHGs worldwide, so minimizing them is a significant way to make an event more sustainable..

Line 580: ...hosting it virtually or having ...

Line 583: In addition, holding events outside or ...

Line 596: ..to produce more visibility through the checklis...

Line 602: This affirms continuous Indigenous presence and rights, acknowledges the ongoing effects of settler colonization, and supports Indigenous people’s political, ...

Line 614: This section was added because the office knows this list is not all-inclusive, and there are..

Line 621:  Creating the document took longer than expected, as using InDesign was a skill everyone in the office needed to improve.....

Line 630: ...and an incentive for event...

Line 634:  ...as its program with its icon. ..

Line 635:  Because event planning happens across campus, the icon's design needed to be TAMU centred...

Line 658: A list of all the Bryan/College Station restaurants was generated..

Line 660: The history of each of the restaurants ..

Line 664: Some restaurants are not technically local but focus on sustainability in their core values...

Line 668: ...the following resources...

Line 674: It seemed obvious to look at Leadership in Energy and Environmental Design (LEED) certified buildings, but only a few are on campus. Multiple buildings on campus are built to LEED standards
but are not certified.

Line 154: ...businesses that could help ...

Line 160: ....also arises at place-based events: the influx of tourists,...

Line 163:Social interaction and favourable relationships between visitors and local residents can generate a welcoming atmosphere and helps to promote local culture and traditions [27]. However, qualitative research by Wilmink-Thomas (2021) also shows, among other things, how consumer demand for sustainable festivals is assumed to change: Either visitors will expect more due to growing awareness and a great time to get informed–or the desire to celebrate rompishly...

Line 682: The downside is that those restaurants close...

Line 685: ...and significant changes ....

Line 688: One crucial aspect of a...

Line 691: The recycling and
composting event signage colours are pulled from the campus's standard recycling branding....

Figure 6. Sample...

Please write at least one sentence below the figure.

Line 697: The following steps for the checklist were to decide on incentives and how to promote the proper certification promotion....

Line 699: significant incentives for certifying...

Line 700: ..partner in the marketing and promoting of the sustainable ...

Line 702: The word would be spread to thousands of campus members through the ASA newsletters, promotion on the OS website, and on OS social media channels....

Line 708: ...so an initial event audit ...

Line 717: ..and watching live was witnessing ...

Line 719: Unfortunately, this was near the end of the semester, and summer break decreased the number of events happening around campus. Therefore, there were few events left to keep any momentum going.....

Line 731:  ...is the feedback the event planner ...

Line 736: ...but would fall under..

Line 744: This submission showed immediately that there needed to be a description of the event on the first page as there needed to be a way to know what the event was otherwise. T

Line 763: 4.4.4. Feedback....move it on the next page.

Line 765: First was an Instructional Assistant Professor who taught event management and operations, examined the draft framework, and offered feedback: • One of the things I did point out was the importance of balance....

Line 767: While I know you would like the events to do as many of these things as possible, some are mutually exclusive within a category and others, in combination with other choices, make for traditionally poor event outcomes. ...

Line 775: ithout actually seeing it applied to an event, my first thought is that your percentages are likely good, though 80% seems quite high, based on what I stated above...

Line 793: The examples below illustrate how event management students at the Research 1 institution in Texas, USA, have begun to assist local event organizers (their clients) in contemplating and implementing micro-level sustainability measures....

Line 795:  Funds provided by a small, internal university grant incentivized the students to implement aspects of the new SEC at the onsite events they planned and implemented for the campus or community client...

Line 798: ...environmental and social criteria...

Line 804: ...must be a more significant factor in future ....

Line 810: The examples below show student....

Line 820: Cooperative learning and collaborative knowledge sharing were fundamental principles...

Line 834: .....contributing to social sustainability.

Line 840: ...

Line 836: The client requested Texas barbequedinner to support their ...

Line 842: One is a “tailgate,” or an outdoor casual picnic that can be catered and used as a way to gather with friends and fans of a sports team before the match. The event management students in the Fall of 2021 planned one such tailgate. ..

Line 853: ...and guests could take home...

Line 856: ...intentionality to support sustainability...

Line 861: As the Spring of 2022 arrived, several student event groups used sustainable event funding to purchase higher-cost and higher-quality items that support the environment physically and socially.

Line 868: hey had planned to arrange cut flowers and discard them after the event....

Line 869: offered sustainable event funding...

Line 884:  ...functioned differently...

 Line 899: ...funding for this iteration ...

Line 900: Again, these are encouraging signs that event clients will come to see what more is possible, get in the habit of making those changes, and ..

Line 906: This is traditionally hosted in person ...

Line 909: The Sustainability Champion Awards aims to recognize and reward individuals - students, faculty, staff, and team members - who have demonstrated exemplary effort, dedication, and leadership throughout the year in fostering a campus culture of sustainability. .

Line 919: The only paper present at the even...

Line 1093: Significant shifts occurred in the event management ...

Line 1094: ...approaches, considerations and literacies ....

Line 1113: ...therefore, event managers must be capable of engaging in collaborative learning and adaptive planning,....

Line 1125: Collaboration, adaptation, continuous learning, and skill building (including soft skills) undertaken in
the sustainable events domain contribute to overall place resilience and preparedness to handle emergent challenges and threats. ..

Line 1136:..An ethic of collaborative engagement among stakeholders facilitates knowledge sharing and continuous learning, helping spur innovation and action in the event domain and sustain the broader destination as it builds
resilience and essential qualities for preparedness to face future threats and issues

LIne 1141: In conclusion, collaborative learning, engagement and continuous improvement are crucial for event-based destinations to respond swiftly to emergent challenges and wicked problems like climate change.

Line 1149: Attention should be paid to difficult-to-measure and, therefore, easy-to-miss aspects, such as building soft skills and practical knowledge ......

Author Response

Dear Authors,

your paper is very interesting and I would like to use it (with your permission)  in my Event Management classes once it has been published!

OUR RESPONSE: We thank Reviewer 3 for the above and for the extensive writing-related edits below. We have incorporated your suggested edits; they have indeed strengthened our manuscript’s readability and contributions. It is our sincere hope that it will benefit students in your Event Management classes.

To improve your paper, please consider following suggestions:

Abstract: This paper aims to share post-pandemic lessons for destination resilience and sustaining the ability of events. It offers a new perspective that reimagines the space and place of events as learning  destinations [1] enmeshed in complex systems. Complexity arises due to the interactions and interrelationships between numerous stakeholders, activities and events in the social-ecological destination system where boundaries are porous, and issues and actions from afar can impact the local. The case presented here describes the micro-level activities and actions undertaken to engage with destination resilience and sustainable event management and certification at a learning destination in Texas, USA. These situated efforts are shown (i) at the campus-wide level for the university and (ii) 

with the collaborative, learning-oriented activities undertaken by students in the event management classes to pilot test the Sustainable Event Certification Checklist that was developed. They

corroborate the general characteristics and criteria of the complex learning destination summarized 

in the paper, along with identifying and discussing the skills, literacies, and lessons learned to 

advance destination resilience and the sustainability of events. Participants in the learning destination draw on practical knowledge and develop soft skills to engage in adaptive planning proactively and collaboratively with other stakeholders to address emergent challenges and practical ....

Line 34: Events Tourism is one of the fastest growing sectors worldwide, offering considerable economic and social benefits locally and globally through........

Line 53: Split the sentence:

Research on the pandemic’s impacts is still in the early stages. However, greater attention will be needed to manage events in the context of infectious disease outbreaks as online to offline life resumes. However, online and virtual activities will continue to be essential aspects of event planning and marketing [8].

Line 60: They also note the need for more attention synergizing wider social, community, and individual resilience perspectives.

Line 70: A collaborative approach to learning how to address the challenges is needed. Every stakeholder can only resolve the complex issues and wicked problems that..

Line 75: Complexity arises due to the interactions and interrelationships. between numerous stakeholders, activities and events in the destination system where boundaries are porous. Consequently, issues and actions from afar can impact the locals.

Line 82: ...place-based approach to post...

Line 86: ...provide a practical case example...

Line 96:...It is a public university tasked with the public good and well-being. It is an educational and recreational destination, hosting numerous sports and social and cultural events. Actions and initiatives towards destination...

Line 107: The following section...

Line 109: ...which sets ...

Line 115:... the loss of tourism...

Line 118: ...and the use of marketing...

Line 120: Governments stepped in to help the event sector worldwide, recognizing its vital importance in sustaining local and regional economies and its social and psychological health values as social isolation progressed in 2020 and 2021.

Line 122:  Outdoor events received government support in many destinations, recognizing their importance in offering economic and social resilience and their potential to become super-spreading sources of infection and outbreaks [3]. In addition, financial and policy support was vital to enable proper guidelines and measures to address crowding, social distancing, masking, etc., as well as developing new marketing....

Line 129: However, numerous other issues and impacts arise in enacting and managing events.

Line 136: ....environmental pollution....

Line 136: Transportation is critical in infrastructure considerations, and its environmental impacts are significant.

Line 147: ..Olympic Games shows..

Line 171: Key stakeholders ranging from local businesses, residents, visitors, non-profit organizations, and various public sector participants engage collaboratively to assist in greening events, sustaining the destination, and facilitating place resilience, including its inhabitants' well-being..

Line 176: ...play key roles in facilitating learning..

Line 188: ...addresses local as well as global issues...

Line 191:  ...short-duration festivals from...

Line 196: They are also events fulfilling important...

Line 199: In addition, as Mair and Laing (2012) note, greening events can be an excellent opportunity to educate and change attitudes and behavior within the organisation and towards the natural environment .

Line 208: ....as part of a more extensive campaign...

Line 209: ...in the UK pride themselves on reaching out to ...

Line 217: ...economy is a green and healthy economy where sustainable development and well-being are achievable..

Line 2222: Applying the circular economy...

Line 230: ...for the conservation of local..

Line 233: ...grows the local community; ...

Line 234: ...with neutral carbon initiatives, serving foods that are fair trade and organic,...

Line 238: ...and issues in the event management...

Line 239: Building sustainability and resilience in the event-based destination also requires learning, adapting, and managing the multiple stakeholders and dynamic context in which issues arise...

Line 259: Attractions like events are not isolated; they are part of the complex destination system. Their sustainability and resilience are interrelated with the emergent and dynamic properties of the complex system and the situated practices of multiple stakeholders within the local system external to it...

Line 262: Therefore, strategies for sustainable event management should be tailored to the place rather than aspire towards universal norms, as Dredge and Whitford (2010) note...

Line 276: Greening is about progress, not perfection. Ahmad et al. (2013) note that sustainability does not happen with just a single event [29]. ...

Line 275: It also requires making trade-offs and decisions that may only check some of the boxes in a sustainability framework....

Line 277: ...action presents opportunities ...

Line 289: The professionals interviewed also
pointed to the need for practical skills (applied knowledge) and stronger cooperation between universities and enterprises [42].

Line 296:  Some important steps they describe are vital stakeholders coming together to identify objectives...

Line 299: ...development of an event..

Line 303: Stakeholder involvement is crucial in planning...

Line 305: events and developing policies...

Line 306: As noted earlier, visitors are also important stakeholders and should be involved through awareness raising and engagement in event sustainability. Wong et al. (2015) examined green policies and...

Line 311:  ...and a healthy environment. ...

Line 319:...the computerization and ...

Line 321: Such virtual adaptations are integral to the post-COVID paradigm shift to facilitate event sustainability and destination resilience. These authors advocate creating new and specific international standards for managing event competition sites in the event of a new pandemic..

Line 329: Surprisingly, despite its immense local to global significance and importance in the context of recreation, leisure and tourism, sustainability-related accreditation schemes for the event industry still need to catch up.

Line 334: However, certifications for individual professionals in sustainable event planning have been gaining traction (see globalgreenevents.org and https://www.mpi.org/education/certificate-programs/sustainable-event-strategist). ...

Line 341: Furthermore, different green levels and logos can also be created, so that the audience can identify the ecological level that the event obtained [47]....

Line 359: The following section...

Line 371: AMU is a destination for student learning and professional development and a hub for numerous recreational activities, including sports, cultural, and community event destinations...

Line 376: ..and is home to men’s and women’s basketball and volleyball teams....

Line 388: The university is an essential focal point for education, recreation, and entertainment, with a significant environmental impact...

Line 390: and lower greenhouse gas emissions.

Line 390: The campus Sustainability Master Plan (SMP) provides a blueprint to make measurable, attainable actions for the university. It focuses on nine themes and defines 16 evergreen goals and 47 targets over a 20-year timeline.

Line 396: TAMU's primary Greenhouse Gas reduction effort is to reduce the overall campus energy consumption.

Line 401: ..size of over 29.6 million..

Line 403: an aggressive building-level energy reduction program known as the ..

Line 407: ..and was an example of how sports..

Line 415: A new baseline should be calibrated regularly to encourage good stewardship. Building systems require constant monitoring. Otherwise, the behaviors..

Line 419: Table 1. The reduction...

Line 424: Communicating with the public about the need for and results of important GHG reduction work is vital in raising awareness. ..

Line 427: ...sustainability of sports initiatives...

Line 465: ..socially sustainable events around campus.

Line 468: With the number of events happening all over campus, many environmental, economic, and social impacts could be minimized or mitigated with ...

Line 470: Reviewing other schools across the nation, many had some sustainable event certification, and it confirmed that our university needed one, too.

Line 473: of greening events as sustainable learning destinations.

Line 480: The first development phase was to..

Line 481: Most people think of the Waste Minimization checklist items when they consider sustainable choices.

Line 482: Reminders to have recycling bins available, use digital and social promotion instead of paper to advertise the event, and source reusable..

Line 486: ..were the following checklist ...

Line 487: ...organizations with limited ..

Line 510: ..actions in the initial ...

Line 512: the event's purpose...

Line 515: The first three items a planner can review are determining whether the event will be promoted digitally and accessible through social media, bulk emails, and campus TV signage.

Line 519: This checklist encourages using reusable promotion forms such as sandwich boards, yard signs, flags, or bus ads that do not have date-specific information...

Line 524:  ...for events have the potential for significant sustainability impact, positive or negative...

Line 525: Offering vegetarian and vegan entree choices is environmentally friendly and includes dietary restrictions.

Line 529: It lets attendees choose what goes on their plates, which can minimize food waste.

Line 538: Due to limited transport and fertiliser usage, serving local and in-season produce has a much smaller environmental impact than imported or out-of-season produce.

Line 542: ...organizations with limited funding sources,...

Line 544: Second, supporting local and sustainable companies is a great way to exercise both environmental
and social responsibility, limiting transportation emissions and providing income to local community members.....

Line 58: ...their cleaning products...

Line 551: Crafting tablescapes and centrepieces from natural items can be inexpensive to create a welcoming look at banquets.

Line 560: ...or other promotional items...

Line 563: After an event, it is customary ...

Line 570:  ..reduce the waste heading to a landfill,..

Line 571: ..important, so attendees...

Line 578: Transportation emissions account for a large amount of GHGs worldwide, so minimizing them is a significant way to make an event more sustainable..

Line 580: ...hosting it virtually or having ...

Line 583: In addition, holding events outside or ...

Line 596: ..to produce more visibility through the checklis...

Line 602: This affirms continuous Indigenous presence and rights, acknowledges the ongoing effects of settler colonization, and supports Indigenous people’s political, ...

Line 614: This section was added because the office knows this list is not all-inclusive, and there are..

Line 621:  Creating the document took longer than expected, as using InDesign was a skill everyone in the office needed to improve.....

Line 630: ...and an incentive for event...

Line 634:  ...as its program with its icon. ..

Line 635:  Because event planning happens across campus, the icon's design needed to be TAMU centred...

Line 658: A list of all the Bryan/College Station restaurants was generated..

Line 660: The history of each of the restaurants ..

Line 664: Some restaurants are not technically local but focus on sustainability in their core values...

Line 668: ...the following resources...

Line 674: It seemed obvious to look at Leadership in Energy and Environmental Design (LEED) certified buildings, but only a few are on campus. Multiple buildings on campus are built to LEED standards
but are not certified.

Line 154: ...businesses that could help ...

Line 160: ....also arises at place-based events: the influx of tourists,...

Line 163:Social interaction and favourable relationships between visitors and local residents can generate a welcoming atmosphere and helps to promote local culture and traditions [27]. However, qualitative research by Wilmink-Thomas (2021) also shows, among other things, how consumer demand for sustainable festivals is assumed to change: Either visitors will expect more due to growing awareness and a great time to get informed–or the desire to celebrate rompishly...

Line 682: The downside is that those restaurants close...

Line 685: ...and significant changes ....

Line 688: One crucial aspect of a...

Line 691: The recycling and
composting event signage colours are pulled from the campus's standard recycling branding....

Figure 6. Sample...

Please write at least one sentence below the figure.

Line 697: The following steps for the checklist were to decide on incentives and how to promote the proper certification promotion....

Line 699: significant incentives for certifying...

Line 700: ..partner in the marketing and promoting of the sustainable ...

Line 702: The word would be spread to thousands of campus members through the ASA newsletters, promotion on the OS website, and on OS social media channels....

Line 708: ...so an initial event audit ...

Line 717: ..and watching live was witnessing ...

Line 719: Unfortunately, this was near the end of the semester, and summer break decreased the number of events happening around campus. Therefore, there were few events left to keep any momentum going.....

Line 731:  ...is the feedback the event planner ...

Line 736: ...but would fall under..

Line 744: This submission showed immediately that there needed to be a description of the event on the first page as there needed to be a way to know what the event was otherwise. T

Line 763: 4.4.4. Feedback....move it on the next page.

Line 765: First was an Instructional Assistant Professor who taught event management and operations, examined the draft framework, and offered feedback: • One of the things I did point out was the importance of balance....

Line 767: While I know you would like the events to do as many of these things as possible, some are mutually exclusive within a category and others, in combination with other choices, make for traditionally poor event outcomes. ...

Line 775: ithout actually seeing it applied to an event, my first thought is that your percentages are likely good, though 80% seems quite high, based on what I stated above...

Line 793: The examples below illustrate how event management students at the Research 1 institution in Texas, USA, have begun to assist local event organizers (their clients) in contemplating and implementing micro-level sustainability measures....

Line 795:  Funds provided by a small, internal university grant incentivized the students to implement aspects of the new SEC at the onsite events they planned and implemented for the campus or community client...

Line 798: ...environmental and social criteria...

Line 804: ...must be a more significant factor in future ....

Line 810: The examples below show student....

Line 820: Cooperative learning and collaborative knowledge sharing were fundamental principles...

Line 834: .....contributing to social sustainability.

Line 840: ...

Line 836: The client requested Texas barbequedinner to support their ...

Line 842: One is a “tailgate,” or an outdoor casual picnic that can be catered and used as a way to gather with friends and fans of a sports team before the match. The event management students in the Fall of 2021 planned one such tailgate. ..

Line 853: ...and guests could take home...

Line 856: ...intentionality to support sustainability...

Line 861: As the Spring of 2022 arrived, several student event groups used sustainable event funding to purchase higher-cost and higher-quality items that support the environment physically and socially.

Line 868: hey had planned to arrange cut flowers and discard them after the event....

Line 869: offered sustainable event funding...

Line 884:  ...functioned differently...

 Line 899: ...funding for this iteration ...

Line 900: Again, these are encouraging signs that event clients will come to see what more is possible, get in the habit of making those changes, and ..

Line 906: This is traditionally hosted in person ...

Line 909: The Sustainability Champion Awards aims to recognize and reward individuals - students, faculty, staff, and team members - who have demonstrated exemplary effort, dedication, and leadership throughout the year in fostering a campus culture of sustainability. .

Line 919: The only paper present at the even...

Line 1093: Significant shifts occurred in the event management ...

Line 1094: ...approaches, considerations and literacies ....

Line 1113: ...therefore, event managers must be capable of engaging in collaborative learning and adaptive planning,....

Line 1125: Collaboration, adaptation, continuous learning, and skill building (including soft skills) undertaken in
the sustainable events domain contribute to overall place resilience and preparedness to handle emergent challenges and threats. ..

Line 1136:..An ethic of collaborative engagement among stakeholders facilitates knowledge sharing and continuous learning, helping spur innovation and action in the event domain and sustain the broader destination as it builds
resilience and essential qualities for preparedness to face future threats and issues

LIne 1141: In conclusion, collaborative learning, engagement and continuous improvement are crucial for event-based destinations to respond swiftly to emergent challenges and wicked problems like climate change.

Line 1149: Attention should be paid to difficult-to-measure and, therefore, easy-to-miss aspects, such as building soft skills and practical knowledge ......

Round 2

Reviewer 2 Report

ikona Zweryfikowane przez społeczność Thank you for the corrections you made. I accept and recommend publication.